# New Panx-1 Blockers: Synthesis, Biological Evaluation and Molecular Dynamic Studies

**DOI:** 10.3390/ijms23094827

**Published:** 2022-04-27

**Authors:** Letizia Crocetti, Gabriella Guerrini, Maria Paola Giovannoni, Fabrizio Melani, Silvia Lamanna, Lorenzo Di Cesare Mannelli, Elena Lucarini, Carla Ghelardini, Junjie Wang, Gerhard Dahl

**Affiliations:** 1NEUROFARBA, Pharmaceutical and Nutraceutical Section, University of Florence, Via Ugo Schiff 6, 50019 Sesto Fiorentino, Italy; letizia.crocetti@unifi.it (L.C.); gabriella.guerrini@unifi.it (G.G.); fabrizio.melani@unifi.it (F.M.); silvia.lamanna@stud.unifi.it (S.L.); 2NEUROFARBA, Pharmacology and Toxicology Section, University of Florence, Viale Pieraccini 6, 50139 Florence, Italy; lorenzo.mannelli@unifi.it (L.D.C.M.); elena.lucarini@unifi.it (E.L.); carla.ghelardini@unifi.it (C.G.); 3Department of Physiology and Biophysics, University of Miami School of Medicine, 1600 N.W. 10th Avenue, Miami, FL 33136, USA; jwang1@miami.edu (J.W.); gdahl@miami.edu (G.D.)

**Keywords:** pannexin, panx-1 blockers, organic synthesis, electrophysiological assay, neuropathic pain, proximity frequencies

## Abstract

The channel protein Panx-1 is involved in some pathologies, such as epilepsy, ischemic stroke, cancer and Parkinson’s disease, as well as in neuropathic pain. These observations make Panx-1 an interesting biological target. We previously published some potent indole derivatives as Panx-1 blockers, and as continuation of the research in this field we report here the studies on additional chemical scaffolds, naphthalene and pyrazole, appropriately substituted with those functions that gave the best results as in our indole series (sulphonamide functions and one/two carboxylic groups) and in Panx-1 blockers reported in the literature (sulphonic acid). Compounds **4** and **13**, the latter being an analogue of the drug Probenecid, are the most potent Panx-1 blockers obtained in this study, with I = 97% and I = 93.7% at 50 µM, respectively. Both compounds, tested in a mouse model of oxaliplatin-induced neuropathic pain, showed a similar anti-hypersensitivity profile and are able to significantly increase the mouse pain threshold 45 min after the injection of the doses of 1 nmol and 3 nmol. Finally, the molecular dynamic studies and the PCA analysis have made it possible to identify a discriminating factor able to separate active compounds from inactive ones.

## 1. Introduction

Pannexins (Panxs) are a family protein forming non-junctional plasma membrane channels and they share similar structural properties with Connexins (Cxs) and Innexins (Inxs), which are gap-junction proteins, putting adjacent cells in communication [1]. Three Panx proteins have been identified, denoted as Panx-1, Panx-2, and Panx-3, with Panx-1 being the only one well characterised [2]. Panxs are generally found throughout the body and their expression within tissues is often cell type-specific [3]; Panx-1 is the most abundant and ubiquitous isoform. Panx1 channels can be activated by different physiological and pathological stimuli, including mechanical stress, increased extracellular K^+^ concentration or increased intracellular Ca^2+^ level, proteolytic cleavage (by Caspase 3 or 7), pro-inflammatory processes and changes in extracellular pH [4,5]. The role of the Panx1 channel in the ATP efflux/release, widely accepted by different research groups since its discovery in 2000s, has recently been called into question because of the channel pore dimension. All the recently solved Panx-1 structures [6,7,8] reveal that the external restriction of the channel has a diameter of about 9Å, which permits the passage of chloride ions but not of ATP, since it requests a greater pore size [9,10]. This aspect will undoubtedly have to be explored further.

The first studies of electron microscopy (EM) of cell membranes supported the hypothesis that Panxs oligomerise into hexameric structures called pannexons [4,11], but the recent cryo-EM studies unambiguously established that Panx-1 channel is an arrangement of seven identical protomers (heptameric channels) [6,12]. From a structural point of view, each subunit consists of a transmembrane domain with four membrane-spanning helices (TM1-4) and folded extracellular and intracellular domains with both the N- and C-terminal domain (C-terminal helix, CHT and N-terminal helix, NHT) within the intracellular space; NHT of a protomer that lined the TMD domain of the adjacent one with a potential role in the assembly of the channel [1,7]. The fourteen extracellular domains (ECDs) were organised into a cap structure, forming the extracellular entrance to the transmembrane pore. Each chain includes tryptophan (Trp74, W74), arginine (Arg75, R75) and aspartic acid (Asp81, D81) residues that interact with each other through cation-π interaction and a salt bridge, and it can potentially operate as a gate containing positive charges. The study conducted by Michalski et al. [8] hypothesised that R75 might be a major determinant of anion selectivity of Panx-1 channels in the open state. The mutation hPanx-1 Arg75Ala (R75A) removed the positive charge and decreased chloride ion selectivity, supporting the idea that the positively charged Arg75 plays a role in the anion selectivity of Panx-1. Again, the simultaneous mutation W74A and R75A abolished the CBX Panx-1 inhibition, reinforcing the critical role of these residues in channel pharmacology and protein–protein interactions. Site-directed mutagenesis experiments have demonstrated that four conserved extracellular cysteine residues are required for functional Panx-1 channel formation [11,13]. Human Panxs are glycosylated at the residue N255 in the extracellular loop (N254 in rat and mouse) and the enzymatic removal of bulky N-glycans from the cell surface or the mutation of N255Q–N255A, enhances the coupling ability of Panxs-1, thus highlighting the role of N255 in membrane trafficking for Panx-1 channel [14]. These discoveries represent a breakthrough in understanding the architecture of Panxs previously unknown and can open to further interpretations of their physiological/pathological role and their unique channel properties.

Recently, the involvement of Panx-1 in some pathologies, such as epilepsy, ischemic stroke, cancer, and Parkinson’s disease has demonstrated [15]. An interesting therapeutic application is neuropathic pain, where the role of Panx-1 channels seems closely related to the purinergic system [16]. Currently, the number of Panx-1 channel blockers available for the study and the comprehension of physiological/pathological role of this protein channel is very limited and, in most cases, the drugs have very different chemical structures and different primary targets. Drugs such as probenecid [17], carbenoxolone (CBX) [18], mefloquine [19], glyburide [20], or compounds such as the food dye Brilliant Blue FCF [21], 5-nitro-2-(3-phenylpropylamino)benzoic acid (NPPB) [17], and the synthetic peptide ^10^Panx, that mimics the sequence of Panx1 in the second extracellular loop area [22], are the most used reference compounds. In Figure 1, some representative products are shown.

As result of an extensive screening of several different scaffold molecules, our team was the first to publish a medicinal chemistry research on Panx-1 channel blockers [23]. Compounds with indole ring nucleus yielded good and encouraging results. Of particular interest are products bearing a secondary and tertiary sulfonamide function at position 5 of the scaffold and one/two carboxylic groups (I% = 100 at 50 µM). The most potent compound **15b** (numbering in the original paper) reported in Figure 1, showed a IC_50_ = 0.8 µM, and in vivo tested in a model of oxaliplatin-induced neuropathy, was able to completely revert the hypersensitivity at the dose of 1 nmol intratecally, thus suggesting a relationship among this effect and the channel blocker potency [23].

As continuation of the research in this field we planned to investigate additional chemical scaffolds, and in the present paper we report the synthesis and biological evaluation of a new series of naphthalene (Figure 2A) and pyrazole (Figure 2B) derivatives bearing those functions that in the previous indoles series gave the best results [23], such as sulphonamide functions and one/two carboxylic groups; moreover, we synthesised sulphonic acid derivatives according to Panx-1 blockers reported in the literature. Finally, we planned to synthesise an analogue of the drug Probenecid bearing two additional acid groups (Figure 2C). All new products were screened as Panx-1 blockers at the dose of 50 µM using the Voltage Clamp technique on *Xenopus laevis* oocytes expressing Panx-1 channels and two selected compounds were tested in a mouse model of oxaliplatin-induced neuropathic pain. Docking experiments using a cryo-EM structure of hPanx-1 channel (PDB 7DWB) on some active and inactive compounds were finally performed with the aim to explain the different activities.

## 2. Results and Discussion

### 2.1. Chemistry

The procedures followed to obtain all the new products are reported in Figure 1, Figure 2, Figure 3, Figure 4, Figure 5 and Figure 6 and the structures are confirmed on the basis of analytical and spectral data. Figure 1 outlines the synthetic pathway affording the final compounds **3a**–**f**, **4** and **5**. The 5-aminonaphthalene-2-sulfonic acid (**1**) and the appropriate aryl bromide of type **2** (both commercially available) were reacted with anhydrous sodium carbonate (Na_2_CO_3_) in dry dimethylformamide (DMF) at reflux for compound **3a** and with sodium hydride in dry DMF at 90 °C for **3b**–**f** [24]. The ester **3f** was then reacted with sodium hydroxide (40%) affording the acid **4**, while the treatment of **3a** with methyl iodide led to compound **5**.

Figure 2 depicts the synthesis of the final compounds **7a**–**g**, the sodium salt of **7c**, compound **9**, and **8** [25]. Starting from the 5-aminonaphthalene-2-sulfonic acid 1, the acylation reaction using the appropriate acids **6a**–**g** (**6a** [26], **6b** [27], **6c** [28], **6d,f** [29], **6g** [30] HOBt, NEt_3_, and DCC in dry DMF led to compounds **7a**–**g**, respectively. The intermediate **6e** was obtained from the phenylsulphonamide **10** [31] by alkylation with 2-phenylethyl bromide **2b** in the standard conditions, and the next alkaline hydrolysis afforded the reagent **6e**. The compound **8** was synthesised from 1 using benzoyl chloride as reported in the reference [25], while **7c**, when treated with fresh sodium ethylate, gave the corresponding sodium salt **9**.

Figure 3 shows the synthesis of the final sulphonamide **13**, analog of the drug Probenecid. The ester **12**, obtained by reacting **10** and methyl 4-(bromomethyl)benzoate **2f** [32], was subjected to alkaline hydrolysis with sodium hydroxide 40% and EtOH 96% at reflux to obtain **13**.

The syntheses of pyrazole derivatives are depicted in Figure 4, Figure 5 and Figure 6. The ethyl 5-amino-1-(4-sulfamoylphenyl)-1H-pyrazole-4-carboxylate **14** [33] was subjected to alkylation with the appropriate alkyl halides in standard conditions, affording the secondary sulphonamides **15a**–**c** which, together with the ester **14**, were transformed into the corresponding acids, **18a**–**c** and **17**, respectively. The secondary sulphonamides **15a** and **15b** were then treated with an excess of the appropriate alkyl halide (1-iodoethane for **15a** and 1-bromopropane for **15b**) furnishing compound **16a**, and a mixture of **16b** (di-alkylated) and **16c** (tri-alkylated), respectively; all compounds of type **16** were finally subjected to alkaline hydrolysis providing the final acids **19a**–**c** (Figure 4).

Instead, the alkylation of the same starting compound **14** with (bromomethyl)benzene led to a mixture of three products as reported in Figure 5: the secondary sulphonamide **20**, the tertiary sulphonamide **21**, and the tertiary sulphonamide **22** bearing also benzylated at the amino group at position 5. Compounds were separated by column chromatography, and **20** and **21** gave the corresponding acids **23** and **24** by alkaline hydrolysis.

A further different trend was recorded by reacting **14** with (2-bromoethyl)benzene; infact this reaction afforded a mixture of two secondary sulphonamides (**25**) and (**26**), one of which has an additional ethyl-benzene group at the amino at position 5 (**26**). The primary sulphonamide ethyl 5-amino-1-[4-(N-phenethylsulfamoyl) phenyl]-1H-pyrazole-4-carboxylate (**25**) was further alkylated with (bromomethyl)benzene affording the secondary sulphonamide **27**. The esters **25**–**27** were then hydrolyzed to the corresponding acids obtaining **28**–**30**, respectively (Figure 6).

### 2.2. Biological Evaluation

All new products were screened as Panx-1 blockers at the dose of 50 µM using the Voltage Clamp technique on *Xenopus Laevis* oocytes expressing Panx-1 channels. The membrane potential of the cells was held at -60 mV and voltage steps to +60 mV were applied to induce membrane currents. To verify that Panx-1 channels carried the currents, in each experiment 100 µM carbenoxolone (CBX) was applied. At this concentration, CBX is known to yield 100% inhibition of Panx-1 activity [4]. If present, leak currents resistant to CBX were subtracted from the total current to assess % inhibition mediated by the tested compounds. Table 1 and Table 2 report the electrophysiological results of naphthalene-2-sulphonic acid derivatives bearing at position 5 an amino- (**3a**–**e**, **4** and **5**) or amide group (**7a**–**g**, **8** and **9**) and of compound **13**. The data of Table 1, suggest that the lengthening of the alkyl chain between the nitrogen at position 5 and the phenyl ring (*n* = 1–5) in compounds of type **3** positively influenced the blocker potency, going from I = 31.7% for **3a** (*n* = 1) to I = 62% for **3e** (*n* = 5). From this trend stands out compound **3c** (*n* = 3), which is most active of this series with I = 66.7%. While the introduction of a methyl at the N-5 of **3a** afforded a compound (**5**) showing a similar activity to **3a** itself (I = 27.3% versus 31.7%), the introduction of an additional acid group (COOH) at para position on the phenyl ring of **3a** (compound **4**) leads to an increase in efficacy, resulting in inhibition values of 97%. 

Table 2 reports the inhibition values of compounds with an amide function at position 5 and a sulphonamide moiety in para position of the phenyl ring (**7a**–**g**, **8** and **9**). The biological results are appreciable, but do not indicate a precise trend of activity related to the nature of the substituents. For example, among the N-alkyl sulphonamide derivatives **7a**–**c**, the most active compounds are **7a** and **7c** (and its sodium salt **9**), bearing ethyl and butyl groups, respectively (I = 54% for **7a**; I = 67.8% for **7c**; I = 65.3% for **9**); while **7b**, bearing propyl groups, was less potent (I = 34%). On the other hand, the elimination of the sulfamoyl fragment (compound **8**) led to a loss of potency (I = 15%), thus confirming the importance of this function for the activity. Moreover the introduction of alkyl/aryl groups on sulphonamide moiety produced interesting results that in this case appeared to be related to the size of the substituents; in particular, the more bulky compounds **7e** and **7f** exhibited lower and similar inhibition values (I = 58.3% and I = 61.3% at 50 µM, respectively) with respect to the N-dibenzyl derivative **7d,** which showed an inhibition value of 75%. The corresponding secondary sulphonamide **7g** is less potent, in agreement with our previous results [23]. Very intriguing was the potency of compound **13**, an analogue of the drug Probenecid, in which the propyl chains of the sulphonamide had been replaced with 4-carboxybenzyl groups. This modification afforded a potent Panx1 blocker with I = 93.7%; thus, it was more potent than Probenecid, which exhibits I = 29% at 50 µM.

Finally, the biological data reported in Table 3 indicated that the pyrazole ring is not a suitable scaffold for the synthesis of Panx-1 blockers, since all compounds showed percentage inhibition values < 30% with the only exception of product **24**, which exhibited I = 42%.

For some selected compounds belonging to naphtalene series (**3c**, **4**, **7c** and **7d**) and for compound **13**, dose-response curves were estabilished (Figure 3 and Figure 4, see also Appendix A). For reference, CBX was applied to verify that the currents were indeed carried by Panx1 channels and did not represent an unspecific leak. The IC_50_ values are reported in Table 4 and the most potent compound **13**, able to almost completely block Panx-1 channel at 50 µM, exhibited interesting efficacy with IC_50_ = 1.2 µM. On the other hand, the other products (**3c**, **4**, **7c** and **7d**) also show appreciable IC_50_ values (>1.2 µM). Although the IC_50_ values are in the micromolar range, they appear to be of considerable interest because, as reported above, they are still lower or comparable to those of most Panx-1 blockers used to study the channel.

### 2.3. In Vivo Studies

The two selected Panx-1 blockers, **4** and **13,** were tested in a mouse model of oxaliplatin-induced neuropathic pain that involves this channel [16]. Oxaliplatin is a third-generation platinum-based chemotherapeutic agent that, similar to other platinum-based compounds, interferes with the proliferation of tumor cells, leading to cancer cell destruction [34] Oxaliplatin is associated with various side effects and one of the most important ones is the development of neurotoxicity, which significantly limits the patient’s quality of life [35]. Recent studies have highlighted the efficacy of selective Panx1 antagonists in reducing pain threshold evoked by repeated administrations of antineoplastic drug, making Panx1 a possible therapeutic target in reducing chemotherapeutic-induced neuropathic pain [16,23]. In this experiment we tested the anti-hypersensitivity properties of two novel Panx1 antagonists in a mouse model of oxaliplatin-induced thermal allodynia. As shown in Figure 5 in the Cold plate test (thermal non-noxious stimulus), repeated oxaliplatin treatment induced a significant reduction of the time spent by the mouse on a cold surface in comparison to the control group (vehicle + vehicle). This time delay is reported here as licking latency (s). The two compounds were acutely injected intrathecally at 1 nmol and 3 nmol in oxaliplatin-treated animals. Compound **4** evoked a significant enhancement of the pain threshold in a dose dependent-manner starting from 30 min after injection. In particular, the maximum anti-allodynic effect was reached at 45 min with the dose of 3 nmol (Figure 5a). Compound **13** showed a similar anti-hypersensitivity profile and provided a significant increase of the mouse pain threshold 45 min after the injection of the doses of 1 nmol and 3 nmol. The effect recorded was dose-dependent (Figure 5b).

### 2.4. Molecular Modeling

With the aim to identify the amino acids of Panx-1 channel involved in the binding of our compounds, we reported the docking and the molecular dynamic study of the most representative newly synthesised compounds. For this study, a reduced-size Panx-1 channel obtained from PDB was used (PDB code 7DWB) and this reduced structure contains all amino acids within a distance of approximately 2.5 nm from the “CBX binding site” [23].

To understand/rationalise the electrophysiological results and the binding mode of these new naphthalene derivatives, compounds **3e**, **4**, **7d**, and **13** (I = 62%, 97%, 75%, and 94%) and **3a**, **7g**, **8**, **24**, and Probenecid (P) (inactive ones), were chosen to make the virtual screening.

The ligands were placed in the binding site through AUTODOCK 4.2 [36], and the best pose found by AUTODOCK was subjected to a MD simulation (10 ns) using GROMACS v5.1 program [37] (see the experimental section for the details). The best binding energy calculated by AUTODOCK for the considered ligands is shown in the Table 5.

Unfortunately, the difference of binding energy between active and inactive compounds is not statistically significant (*p* > 0.05) and these values do not explain the electrophysiological results.

The analysis of the MD simulation trajectories permitted calculating the most significant ‘Proximity Frequencies’ (PF) [38] of the amino acids, within a distance < 0.23 nm, for at least 50% of the time. From all the PFs collected, those with higher variance (standard deviation > 30) were selected. It is clear that the difference of the sums of PFs, between active and inactive compounds, is significantly different (*p* < 0.05), suggesting that these parameters may be discriminatory.

In Table 6, the sum of most significative PFs (standard deviation > 30) selected for single, double, and triple aminoacids is reported.

The greatest discriminated capacity, between active and inactive compounds, is found when the ligands are simultaneously in proximity to two amino acids (double PF, average *p* = 0.004). From the PCA (Principal Components Analysis) it is evident that the inactive compounds, having a lower number of PFs, are “grouped” in the center of the score plot (low PCs values), while the active compounds are not “grouped” (Figure 6).

The dispersion of active ligands in the score plots suggests that the activity could be explained by the total number of bonds that can be established between ligands and protein instead of the ligand position in the protein. The amino acids found in close proximity to the ligands most frequently (and they are likely involved in forming hydrogen bonds) are Arg75, Ser59 and Ser82. These amino acids belong to different chains and are found in the vicinity of the ligands also in double and triple combinations.

## 3. Materials and Methods

### 3.1. Chemistry

Reagents and starting materials were obtained from commercial sources (Merck Life Science srl, Milan, Italy). Extracts were dried over Na_2_SO_4_, and the solvents were removed under reduced pressure. All reactions were monitored by thin-layer chromatography (TLC) using commercial plates pre-coated with Merck silica gel 60 F-254. Visualisation was performed by UV fluorescence (λ_max_ = 254 nm) or by staining with iodine or potassium permanganate. Chromatographic separations were performed on a silica gel column by gravity chromatography (Kieselgel 40, 0.063–0.200 mm; Merck), flash chromatography (Kieselgel 40, 0.040–0.063 mm; Merck), silica gel preparative TLC (Kieselgel 60 F_254_, 20 × 20 cm, 2 mm). Yields refer to chromatographically and spectroscopically pure compounds, unless otherwise stated. Compounds were named following IUPAC rules, as applied by Beilstein-Institut AutoNom 2000 (4.01.305) or CA Index Name. All melting points were determined on a microscope hot stage Büchi apparatus and are uncorrected. The identity and purity of intermediates and final compounds were ascertained through ^1^H-NMR, ^13^C-NMR, and TLC chromatography. Monodimensional spectra ^1^H-NMR and ^13^C-NMR were registered by a 400 MHz field through Avance 400 apparatus (Bruker Biospin Version 002 with SGU). Chemical shifts (d) are in parts per million (ppm) approximated by the nearest 0.01 ppm, using the solvent as internal standard. Coupling constants (J) are in Hz, they were calculated by Top Spin 3.1 and approximated by 0.1 Hz. Data are reported as follows: chemical shift, multiplicity (exch, exchange; br, broad; s, singlet; d, doublet; t, triplet; q, quartet; m, multiplet; or a combination of those, e.g, dd), integral, assignments, and coupling constant. Mass spectra (*m*/*z*) were recorded on a ESI-MS triple quadrupole (Varian 1200 L) system, in positive ion mode, by infusing a 10 mg/L solution of each analyte dissolved in a mixture of mQ H_2_O:acetonitrile 1:1 *v*/*v*. All new compounds possess a purity ≥ 95%; microanalyses indicated by the symbols of the elements were performed with a Perkine-Elmer 260 elemental analyzer for C, H, and N, and they were within ±0.4% of the theoretical values.

#### 3.1.1. 5-(Benzylamino)naphthalene-2-Sulfonic Acid (**3a**)

A suspension of 0.89 mmol of 5-aminonaphthalene-2-sulfonic acid (**1**), which is commercially available, and 1.79 mmol of Na_2_CO_3_ in 5 mL of dry DMF was stirred at room temperature for 30 min. Then, 1.33 mmol of benzyl bromide (**2a**) were added and the mixture was heated at reflux for 2–3 h. After cooling, the pH was checked and adjusted to 6–7 and the solvent was evaporated. The crude product was purified by flash column chromatography using CH_2_Cl_2_/MeOH/CH_3_COOH 80:20:2 as eluent to obtain the desired compound as white powder [24]. Yield = 86%; mp > 300 °C (EtOH/H_2_O). ^1^H-NMR (400 MHz, DMSO-d_6_) δ 4.48 (s, 2H, CH_2_-Ph), 6.36 (d, 1H, Ar, J = 6.4 Hz), 6.93 (exch br s, 1H, NH), 7.05–7.10 (m, 1H, Ar), 7.15–7.25 (m, 2H, Ar), 7.28–7.35 (m, 2H, Ar), 7.39–7.45 (m, 2H, Ar), 7.60 (d, 1H, Ar, J = 8.8 Hz), 7.95 (s, 1H, Ar), 8.18 (d, 1H, Ar, J = 8.0 Hz). ^13^C-NMR (100 MHz, DMSO-d_6_) δ 48.4 (CH_2_), 119.8 (CH), 121.5 (CH), 123.2 (CH), 125.1 (C), 126.5 (CH), 126.7 (CH), 128.2 (CH), 128.5 (CH), 133.2 (C), 134.1 (C), 139.9 (C), 141.0 (C). ESI-MS calcd. for C_17_H_15_NO_3_S, 313.08; found: *m*/*z* 314.08 [M + H]^+^. Anal. C_17_H_15_NO_3_S (C, H, N).

#### 3.1.2. General Procedure for Synthesis of Compounds **3b**–**f**

A total of 0.90 mmol of 5-aminonaphthalene-2-sulfonic acid (**1**) and 1.80 mmol of sodium hydride (60% dispersion in mineral oil) were dissolved in 5 mL of dry DMF. After 30 min in stirring at room temperature, 1.35 mmol of appropriate phenyllalkyl bromide (**2b**–**f**), which is commercially available, were added and the mixture was stirred at 90 °C for 3–4 h. After cooling, the pH was checked and adjusted to 6–7 and the solvent was evaporated. The crude products were purified by flash column chromatography using CH_2_Cl_2_/MeOH/CH_3_COOH 80:20:2 as eluent.

**5**-(Phenethylamino)naphthalene-2-sulfonic acid (**3b**). Yield = 60%; mp > 300 °C (EtOH/H_2_O). ^1^H-NMR (400 MHz, DMSO-d_6_) δ 2.98 (t, 2H, NHCH_2_*CH_2_*Ph, *J =* 7.6 Hz), 3.40 (q, 2H, NH*CH_2_*CH_2_Ph, *J =* 7.0 Hz), 6.21 (t, 1H, NH, *J =* 4.8 Hz), 6.55 (d, 1H, Ar, *J =* 7.6 Hz), 7.11 (d, 1H, Ar, *J =* 8.0 Hz), 7.17–7.22 (m, 1H, Ar), 7.26–7.32 (m, 5H, Ar), 7.55 (dd, 1H, Ar, *J_1_ =* 8.8 Hz, *J_2_ =* 1.2 Hz), 7.94 (s, 1H, Ar), 8.06 (d, 1H, Ar, *J =* 8.8 Hz). ^13^C-NMR (100 MHz, DMSO-d_6_) δ 34.7 (CH_2_), 45.3 (CH_2_), 104.5 (CH), 116.6 (CH), 121.9 (CH), 121.9 (CH), 123.2 (C), 125.0 (CH), 126.6 (CH), 128.1 (CH), 128.8 (CH), 129.2 (CH), 129.5 (CH), 133.6 (C), 140.4 (C), 144.1 (C), 144.2 (C). ESI-MS calcd. for C_18_H_17_NO_3_S, 327.09; found: *m*/*z* 328.10 [M + H]^+^. Anal. C_18_H_17_NO_3_S (C, H, N).

5-[(3-Phenylpropyl)amino]naphthalene-2-sulfonic acid (**3c**). Yield = 46%; mp > 300 °C (EtOH/H_2_O). ^1^H-NMR (400 MHz, DMSO-d_6_) δ 1.98 (quin, 2H, NHCH_2_*CH_2_*CH_2_Ph, *J =* 7.2 Hz), 2.72 (t, 2H, NHCH_2_CH_2_*CH_2_*Ph, *J =* 7.6 Hz), 3.17 (q, 2H, NH*CH_2_*CH_2_CH_2_Ph, *J =* 6.2 Hz), 6.10 (t, 1H, NH, *J =* 4.8 Hz), 6.42 (d, 1H, Ar, *J =* 7.6 Hz), 7.08 (d, 2H, Ar, *J =* 8.0 Hz), 7.16–7.29 (m, 5H, Ar), 7.54 (d, 1H, Ar, *J =* 8.8 Hz), 7.92 (s, 1H, Ar), 8.09 (d, 1H, Ar, *J =* 8.8 Hz). ^13^C-NMR (100 MHz, DMSO-d_6_) δ 30.4 (CH_2_), 33.4 (CH_2_), 43.3 (CH_2_), 103.8 (CH), 116.3 (CH), 121.8 (CH), 122.1 (CH), 123.1 (C), 125.2 (CH), 126.2 (CH), 127.7 (CH), 128.7 (CH), 128.8 (CH), 133.7 (C), 142.4 (C), 144.6 (C), 145.6 (C). ESI-MS calcd. for C_19_H_19_NO_3_S, 341.11; found: *m*/*z* 342.11 [M + H]^+^. Anal. C_19_H_19_NO_3_S (C, H, N).

5-[(4-Phenylbutyl)amino]naphthalene-2-sulfonic acid (**3d**). Yield = 63%; mp > 300 °C (EtOH/H_2_O). ^1^H-NMR (400 MHz, DMSO-d_6_) δ 1.60–1.70 (m, 4H, NHCH_2_*CH_2_CH_2_*CH_2_Ph), 2.57–2.62 (m, 2H, NHCH_2_CH_2_CH_2_*CH_2_*Ph), 3.10–3.20 (m, 2H, NH*CH_2_*CH_2_CH_2_CH_2_Ph), 6.14 (exch br s, 1H, NH), 6.45 (d, 1H, Ar, *J =* 7.2 Hz), 7.07 (d, 1H, Ar, *J =* 7.6 Hz), 7.12–7.22 (m, 6H, Ar), 7.56 (d, 1H, Ar, *J =* 8.4 Hz), 7.96 (s, 1H, Ar), 8.13 (d, 1H, Ar, *J =* 8.4 Hz). ^13^C-NMR (100 MHz, DMSO-d_6_) δ 28.2 (CH_2_), 29.2 (CH_2_), 35.4 (CH_2_), 43.5 (CH_2_), 103.9 (CH), 116.0 (CH), 121.9 (CH), 122.0 (CH), 123.1 (C), 124.9 (CH), 126.1 (CH), 127.8 (CH), 128.7 (CH), 128.8 (CH), 133.7 (C), 142.6 (C), 144.6 (C), 145.0 (C). ESI-MS calcd. for C_20_H_21_NO_3_S, 355.12; found: *m*/*z* 356.13 [M + H]^+^. Anal. C_20_H_21_NO_3_S (C, H, N).

5-[(5-Phenylpentyl)amino]naphthalene-2-sulfonic acid (**3e**). Yield = 67%; mp > 300 °C (EtOH/H_2_O). ^1^H-NMR (400 MHz, DMSO-d_6_) δ 1.35–1.45 (m, 2H, NHCH_2_CH_2_*CH_2_*CH_2_CH_2_Ph), 1.59 (quin, 2H, NHCH_2_*CH_2_*CH_2_CH_2_CH_2_Ph, *J =* 7.2 Hz), 1.68 (quin, 2H, NHCH_2_CH_2_CH_2_*CH_2_*CH_2_Ph, *J =* 7.2 Hz), 2.55 (t, 2H, NHCH_2_CH_2_CH_2_CH_2_*CH_2_*Ph, *J =* 7.6 Hz), 3.13 (t, 2H, NH*CH_2_*CH_2_CH_2_CH_2_CH_2_Ph, *J =* 7.2 Hz), 6.09 (exch br s, 1H, NH), 6.45 (d, 1H, Ar, *J =* 7.2 Hz), 7.07 (d, 1H, Ar, *J =* 8.0 Hz), 7.10–7.26 (m, 6H, Ar), 7.55 (d, 1H, Ar, *J =* 8.4 Hz), 7.95 (s, 1H, Ar), 8.11 (d, 1H, Ar, *J =* 8.8 Hz). ^13^C-NMR (100 MHz, DMSO-d_6_) δ 26.7 (CH_2_), 28.2 (CH_2_), 31.2 (CH_2_), 35.5 (CH_2_), 43.4 (CH_2_), 104.3 (CH), 116.2 (CH), 121.8 (CH), 122.0 (CH), 123.2 (C), 125.0 (CH), 126.1 (CH), 128.1 (CH), 128.7 (CH), 133.5 (C), 142.7 (C), 144.0 (C), 144.4 (C). ESI-MS calcd. for C_21_H_23_NO_3_S, 369.14; found: *m*/*z* 370.14 [M + H]^+^. Anal. C_21_H_23_NO_3_S (C, H, N).

5-[(4-(Methoxycarbonyl)benzyl)amino]naphthalene-2-sulfonic acid (**3f**). Yield = 20%; mp > 300 °C (EtOH). ^1^H-NMR (400 MHz, DMSO-d_6_) δ 3.79 (s, 3H, O*CH_3_*), 4.54 (d, 2H, NH*CH_2_*Ph, *J =* 5.2 Hz), 6.27 (d, 1H, Ar, *J =* 7.2 Hz), 7.03 (t, 1H, NH, *J =* 5.6 Hz), 7.06–7.13 (m, 1H, Ar), 7.47–7.52 (m, 3H, Ar), 7.60 (d, 1H, Ar, *J =* 8.8 Hz), 7.87 (d, 2H, Ar, *J =* 8.0 Hz), 7.94 (s, 1H, Ar), 8.17 (d, 1H, Ar, *J =* 8.8 Hz). ESI-MS calcd. for C_19_H_17_NO_5_S, 371.41; found: *m*/*z* 372.09 [M + H]^+^. Anal. C_19_H_17_NO_5_S (C, H, N).

#### 3.1.3. 4-[(6-Sulfonaphthalen-1-yl)aminomethyl]benzoic Acid (**4**)

A mixture of 0.20 mmol of compound **3f**, 40% NaOH (3.2 mL) and a little amount of EtOH 96% was stirred at reflux for 3 h. After cooling and maintaining the reaction in an ice bath, HCl 6N was added up to acid pH and the precipitate was filtered off. The acid derivative was purified by flash column chromatography using CH_2_Cl_2_/MeOH/CH_3_COOH 80:20:2 as eluent. Yield = 73%; mp > 300 °C (EtOH/H_2_O). ^1^H-NMR (400 MHz, DMSO-d_6_) δ 4.53 (d, 2H, NHCH_2_Ph, J = 6.0 Hz), 6.22 (d, 1H, Ar, J = 8.4 Hz), 7.26 (t, 1H, NH, J = 5.6 Hz), 7.45 (d, 3H, Ar, J = 8.0 Hz), 7.68 (d, 1H, Ar, J = 8.8 Hz), 7.85 (d, 2H, Ar, J = 8.0 Hz), 8.00 (s, 1H, Ar), 8.25 (d, 2H, Ar, J = 9.6 Hz), 12.76 (exch br s, 1H, COOH). ^13^C-NMR (100 MHz, DMSO-d_6_) δ 48.4 (CH_2_), 119.8 (CH), 123.3 (CH), 125.0 (CH), 126.5 (CH), 126.8 (CH), 128.2 (CH), 128.6 (CH), 130.1 (CH), 133.2 (C), 134.1 (C), 141.1 (C), 145.1 (C), 169.3 (C). ESI-MS calcd. for C_18_H_15_NO_5_S, 357.07; found: *m*/*z* 358.07 [M + H]^+^. Anal. C_18_H_15_NO_5_S (C, H, N).

#### 3.1.4. 5-(Benzyl(methyl)amino)naphthalene-2-sulfonic Acid (**5**)

A total of 0.32 mmol of compound **3a** and 0.64 mmol of sodium hydride (60% dispersion in mineral oil) were dissolved in 4 mL of dry DMF. After 30 min in stirring, 1.28 mmol of methyl iodide was added and the mixture was stirred at 50–60 °C for 5 h. After cooling, ice-cold water (15 mL) was added and the pH was adjusted to 6–7. The suspension was extracted with CH_2_Cl_2_ (3 × 15 mL), dried on sodium sulfate and evaporated. The crude product was purified by flash column chromatography using CH_2_Cl_2_/MeOH/NH_3_ 80:20:2 as eluent. Yield = 95%; mp > 300 °C (EtOH/H_2_O). ^1^H-NMR (400 MHz, DMSO-d_6_) δ 2.70 (s, 3H, CH_3_), 4.24 (s, 2H, CH_2_-Ph), 7.12 (d, 1H, Ar, J = 7.2 Hz), 7.23 (d, 1H, Ar, J = 6.8 Hz), 7.28–7.39 (m, 5H, Ar), 7.60 (d, 1H, Ar, J = 8.0 Hz), 7.69 (d, 1H, Ar, J = 8.8 Hz), 8.08 (s, 1H, Ar), 8.20 (d, 1H, Ar, J = 8.8 Hz). ^13^C-NMR (100 MHz, DMSO-d_6_) δ 42.3 (CH_3_), 58.4 (CH_2_), 116.0 (CH), 124.2 (CH), 124.5 (CH), 125.7 (CH), 127.9 (CH), 127.9 (CH), 128.5 (CH), 131.3 (C), 134.8 (C), 138.7 (C), 141.1 (C), 151.1 (C). ESI-MS calcd. for C_18_H_17_NO_3_S, 327.09; found: *m*/*z* 328.10 [M + H]^+^. Anal. C_18_H_17_NO_3_S (C, H, N).

#### 3.1.5. General Procedure for Synthesis of Compounds **7a**–**g**

A total of 1.31 mmol of acid intermediates **6a**–**g** (6a [26], 6b [27], 6c [28], 6d,f [29], 6g [30]) were dissolved in 13 mL of dry DMF, under N_2_ flow. Then, 1.31 mmol of hydroxybenzotriazole (HOBt), 2.62 mmol of triethylamine and 1.96 mmol of 5-aminonaphthalene-2-sulfonic acid (1) were added. The mixture was maintained in an ice bath and 1.31 mmol of DCC was added. After 15 min in the ice bath, the mixture was heated at 60 °C for 48 h. Finally, after cooling, the solvent was evaporated and the crude products were purified by flash column chromatography using CH_2_Cl_2_/MeOH/CH_3_COOH 90:10:1 then 80:20:2 as eluents.

5-[4-(N,N-diethylsulfamoyl)benzamido]naphthalene-2-sulfonic acid (**7a**). Yield = 20%; mp > 300 °C (EtOH/H_2_O). ^1^H-NMR (400 MHz, DMSO-d_6_) δ 1.05 (t, 6H, N(CH_2_*CH_3_*)_2_, *J =* 7.0 Hz), 3.20 (q, 4H, N(*CH_2_*CH_3_)_2_, *J =* 7.2 Hz), 7.55 (t, 1H, Ar, *J =* 7.8 Hz), 7.61 (d, 1H, Ar, *J =* 6.8 Hz), 7.72 (dd, 1H, Ar, *J_1_ =* 7.2 Hz, *J_2_ =* 1.6 Hz), 7.90–7.96 (m, 4H, Ar), 8.18 (d, 1H, Ar, *J =* 0.8 Hz), 8.24 (d, 2H, Ar, *J =* 8.0 Hz), 10.68 (exch br s, 1H, NH). ^13^C-NMR (100 MHz, DMSO-d_6_) δ 14.6 (CH_3_), 42.4 (CH_2_), 123.5 (CH), 124.5 (CH), 124.8 (CH), 124.9 (CH), 126.4 (CH), 127.3 (CH), 127.6 (CH), 129.2 (C), 129.4 (CH), 133.5 (C), 133.8 (C), 138.5 (C), 142.8 (C), 146.1 (C), 165.6 (C). ESI-MS calcd. for C_21_H_22_N_2_O_6_S_2_, 462.09; found: *m*/*z* 463.10 [M + H]^+^. Anal. C_21_H_22_N_2_O_6_S_2_ (C, H, N).

5-[4-(N,N-dipropylsulfamoyl)benzamido]naphthalene-2-sulfonic acid (**7b**). Yield = 46%; mp > 300 °C (EtOH/H_2_O). ^1^H-NMR (400 MHz, DMSO-d_6_) δ 0.81 (t, 6H, N(CH_2_CH_2_*CH_3_*)_2_, *J =* 7.2 Hz), 1.47 (sex, 4H, N(CH_2_*CH_2_*CH_3_)_2_, *J =* 7.2 Hz), 3.05 (t, 4H, N(*CH_2_*CH_2_CH_3_)_2_, *J =* 7.2 Hz), 7.54 (t, 1H, Ar, *J =* 6.8 Hz), 7.55–7.60 (m, 1H, Ar), 7.70 (d, 1H, Ar, *J =* 8.4 Hz), 7.90–8.00 (m, 4H, Ar), 8.17 (s, 1H, Ar), 8.23 (d, 2H, *J =* 7.6 Hz), 10.65 (exch br s, 1H, NH). ^13^C-NMR (100 MHz, DMSO-d_6_) δ 11.5 (CH_3_), 20.0 (CH_2_), 52.4 (CH_2_), 119.9 (CH), 121.5 (CH), 123.3 (CH), 126.5 (CH), 127.5 (CH), 127.6 (CH), 127.3 (CH), 127.6 (CH), 133.2 (C), 137.5 (C), 140.5 (C), 141.0 (C), 143.8 (C), 146.1 (C), 164.7 (C). ESI-MS calcd. for C_23_H_26_N_2_O_6_S_2_, 490.12; found: *m*/*z* 491.13 [M + H]^+^. Anal. C_23_H_26_N_2_O_6_S_2_ (C, H, N).

5-[4-(N,N-dibutylsulfamoyl)benzamido]naphthalene-2-sulfonic acid (**7c**). Yield = 46%; mp > 300 °C (EtOH/H_2_O). ^1^H-NMR (400 MHz, DMSO-d_6_) δ 0.88 (t, 6H, N(CH_2_CH_2_CH_2_*CH_3_*)_2_, *J =* 7.2 Hz), 1.22–1.31 (m, 4H, N(CH_2_CH_2_*CH_2_*CH_3_)_2_), 1.43–1.50 (m, 4H, N(CH_2_*CH_2_*CH_2_CH_3_)_2_), 3.10 (t, 4H, N(*CH_2_*CH_2_CH_2_CH_3_)_2_, *J =* 7.4 Hz), 7.56 (t, 1H, Ar, *J =* 7.6 Hz), 7.62 (d, 1H, Ar, *J =* 7.2 Hz), 7.72 (d, 1H, Ar, *J =* 8.8 Hz), 7.90–8.00 (m, 4H, Ar), 8.18 (s, 1H, Ar), 8.26 (d, 2H, Ar, *J =* 8.0 Hz), 10.66 (exch br s, 1H, NH). ^13^C-NMR (100 MHz, DMSO-d_6_) δ 14.0 (CH_3_), 19.7 (CH_2_), 30.9 (CH_2_), 48.2 (CH_2_), 123.5 (CH), 124.4 (CH), 124.9 (CH), 126.5 (CH), 127.4 (CH), 127.7 (CH), 129.2 (CH), 129.3 (CH), 133.4 (C), 133.6 (C), 138.4 (C), 142.3 (C), 145.9 (C), 165.6 (C). ESI-MS calcd. for C_25_H_30_N_2_O_6_S_2_, 518.15; found: *m*/*z* 519.16 [M + H]^+^. Anal. C_25_H_30_N_2_O_6_S_2_ (C, H, N).

5-[4-(N,N-dibenzylsulfamoyl)benzamido]naphthalene-2-sulfonic acid (**7d**). Yield = 31%; mp > 300 °C (EtOH/H_2_O). ^1^H-NMR (400 MHz, DMSO-d_6_) δ 4.34 (s, 4H, N(*CH_2_*Ph)_2_), 7.08–7.21 (m, 10H, Ar), 7.56 (t, 1H, Ar, *J =* 7.6 Hz), 7.63 (d, 1H, Ar, *J =* 7.2 Hz), 7.73 (d, 1H, Ar, *J =* 8.4 Hz), 7.94 (t, 2H, Ar, *J =* 9.0 Hz), 8.05 (d, 2H, Ar, *J =* 8.4 Hz), 8.18 (s, 1H, Ar), 8.27 (d, 2H, Ar, *J =* 8.0 Hz), 10.69 (exch br s, 1H, NH). ^13^C-NMR (100 MHz, DMSO-d_6_) δ 51.9 (CH_2_), 123.5 (CH), 124.9 (CH), 126.2 (CH), 126.5 (CH), 127.9 (CH), 128.7 (CH), 129.1 (CH), 129.3 (C), 129.4 (CH), 133.5 (C), 133.9 (C), 136.5 (C), 138.7 (C), 142.4 (C), 146.0 (C), 165.5 (C). ESI-MS calcd. for C_31_H_26_N_2_O_6_S_2_, 586.12; found: *m*/*z* 587.13 [M + H]^+^. Anal. C_31_H_26_N_2_O_6_S_2_ (C, H, N).

5-[4-(N,N-diphenethylsulfamoyl)benzamido]naphthalene-2-sulfonic acid (**7e**). Yield = 20%; mp > 300 °C (EtOH/H_2_O). ^1^H-NMR (400 MHz, DMSO-d_6_) δ 2.78 (t, 4H, N(CH_2_*CH_2_*Ph)_2_, *J =* 7.8 Hz), 3.38 (t, 4H, N(*CH_2_*CH_2_Ph)_2_, *J =* 7.8 Hz), 7.21–7.31 (m, 10H, Ar), 7.54 (t, 1H, Ar, *J =* 7.8 Hz), 7.61 (d, 1H, Ar, *J =* 7.2 Hz), 7.70 (dd, 1H, Ar, *J_1_ =* 8.8 Hz, *J_2_ =* 1.2 Hz), 7.93 (t, 2H, Ar, *J =* 8.8 Hz), 7.99 (d, 2H, Ar, *J =* 8.4 Hz), 8.17 (s, 1H, Ar), 8.24 (d, 2H, Ar, *J =* 8.4 Hz), 10.65 (exch br s, 1H, NH). ^13^C-NMR (100 MHz, DMSO-d_6_) δ 35.3 (CH_2_), 50.1 (CH_2_), 123.4 (CH), 124.5 (CH), 124.7 (CH), 124.8 (CH), 126.4 (CH), 126.9 (CH), 127.5 (CH), 127.6 (CH), 128.9 (CH), 129.2 (C), 129.3 (CH), 129.5 (CH), 133.5 (C), 133.8 (C), 138.7 (C), 138.9 (C), 142.2 (C), 146.3 (C), 165.5 (C). ESI-MS calcd. for C_33_H_30_N_2_O_6_S_2_, 614.15; found: *m*/*z* 615.16 [M + H]^+^. Anal. C_33_H_30_N_2_O_6_S_2_ (C, H, N).

5-[4-(N-benzyl-N-phenethylsulfamoyl)benzamido]naphthalene-2-sulfonic acid (**7f**). Yield = 27%; mp > 300 °C (EtOH/H_2_O). ^1^H-NMR (400 MHz, DMSO-d_6_) δ 2.55 (t, 2H, NCH_2_*CH_2_*Ph, *J =* 8.0 Hz), 3.29 (t, 2H, N*CH_2_*CH_2_Ph, *J =* 8.0 Hz), 4.43 (s, 2H, N*CH_2_*Ph), 7.00 (d, 2H, Ar, *J =* 6.8 Hz), 7.16–7.24 (m, 3H, Ar), 7.30–7.38 (m, 5H, Ar), 7.56 (t, 1H, Ar, *J =* 7.8 Hz), 7.62 (d, 1H, Ar, *J =* 7.2 Hz), 7.72 (d, 1H, Ar, *J =* 8.8 Hz), 7.94 (t, 2H, Ar, *J =* 8.8 Hz), 8.05 (d, 2H, Ar, *J =* 8.4 Hz), 8.18 (s, 1H, Ar), 8.27 (d, 2H, Ar, *J =* 8.0 Hz), 10.70 (exch br s, 1H, NH). ^13^C-NMR (100 MHz, DMSO-d_6_) δ 34.9 (CH_2_), 49.9 (CH_2_), 51.9 (CH_2_), 123.4 (CH), 124.5 (CH), 124.8 (CH), 124.9 (CH), 126.4 (CH), 126.8 (CH), 127.5 (CH), 128.2 (CH), 128.8 (CH), 128.9 (CH), 129.0 (CH), 129.2 (C), 129.5 (CH), 133.5 (C), 133.9 (C), 137.2 (C), 138.6 (C), 138.7 (C), 142.3 (C), 146.2 (C), 165.5 (C). ESI-MS calcd. for C_32_H_28_N_2_O_6_S_2_, 600.14; found: *m*/*z* 601.14 [M + H]^+^. Anal. C_32_H_28_N_2_O_6_S_2_ (C, H, N).

5-[4-(N-phenethylsulfamoyl)benzamido]naphthalene-2-sulfonic acid (**7g**). Yield = 21%; mp > 300 °C (EtOH/H_2_O). ^1^H-NMR (400 MHz, DMSO-d_6_) δ 2.65–2.75 (m, 2H, NCH_2_*CH_2_*Ph), 3.00–3.10 (m, 2H, N*CH_2_*CH_2_Ph), 7.10–7.20 (m, 2H, Ar), 7.27 (d, 2H, Ar, *J =* 6.8 Hz), 7.56 (t, 1H, Ar, *J =* 7.2 Hz), 7.61 (d, 1H, Ar, *J =* 7.8 Hz), 7.71 (d, 2H, SO_2_*NH* + 1H Ar, *J =* 8.0 Hz), 7.90–8.00 (m, 5H, Ar), 8.18 (s, 1H, Ar), 8.23 (d, 2H, Ar, *J =* 6.8 Hz), 10.63 (exch br s, 1H, NH). ^13^C-NMR (100 MHz, DMSO-d_6_) δ 35.8 (CH_2_), 44.6 (CH_2_), 123.4 (CH), 124.5 (CH), 124.6 (CH), 124.8 (CH), 126.4 (CH), 126.7 (CH), 127.1 (CH), 127.6 (CH), 128.8 (CH), 129.1 (CH), 129.2 (CH), 133.5 (C), 133.9 (C), 138.4 (C), 139.1 (C), 143.4 (C), 146.3 (C), 165.6 (C). ESI-MS calcd. for C_25_H_22_N_2_O_6_S_2_, 510.09; found: *m*/*z* 511.10 [M + H]^+^. Anal. C_25_H_22_N_2_O_6_S_2_ (C, H, N).

#### 3.1.6. 5-Benzamidonaphthalene-2-sulfonic Acid (**8**) 

A total of 0.45 mmol of 5-aminonaphthalene-2-sulfonic acid (**1**) was dissolved in water (2 mL) and the pH wad adjusted to 3-4. To the stirred solution, 0.63 mmol of benzoyl chloride dissolved in 1 mL of toluene was slowly added. The reaction mixture was kept at a constant pH of 4 by automatic addition of a 2 M Na_2_CO_3_ solution at room temperature for 6 h. After separating the toluene from the water phase, pH was adjusted to 2.0 with HCl 6N and the aqueous phase was extracted with diethyl ether (4 × 15 mL). The organic phase was removed, the aqueous phase was neutralised (pH = 7) with 2M Na_2_CO_3_ solution and then evaporated under vacuum. The crude product was purified by recrystallisation from methanol to obtain the desired compound [25].

#### 3.1.7. Sodium 5-[4-(N,N-dibutylsulfamoyl)benzamido]naphthalene-2-sulfonate (**9**)

To a fresh solution of sodium ethylate (0.193 mmol of Na° in 1 mL of absolute ethanol), 0.193 mmol of compound **7c** was added. The mixture was stirred at room temperature for 1 h. The solvent was evaporated affording the corresponding sodium salt. Yield = 99%. ^1^H-NMR (400 MHz, D_2_O-d_2_) δ 0.65–0.90 (m, 6H, N(CH_2_CH_2_CH_2_CH_3_)_2_), 1.10–1.29 (m, 4H, N(CH_2_CH_2_CH_2_CH_3_)_2_), 1.30–1.50 (m, 4H, N(CH_2_CH_2_CH_2_CH_3_)_2_), 3.00–3.25 (m, 4H, N(CH_2_CH_2_CH_2_CH_3_)_2_), 7.40–7.60 (m, 2H, Ar), 7.65–7.80 (m, 2H, Ar), 7.80–8.15 (m, 5H, Ar), 8.20–8.40 (m, 1H, Ar). Anal. C_25_H_29_N_2_NaO_6_S_2_ (C, H, N).

#### 3.1.8. General Procedure for Synthesis of Compounds **11** and **12**

A mixture of 1.19 mmol of ethyl 4-sulfamoyl benzoate **10** [31] and 2.38 mmol of anhydrous K_2_CO_3_ in 20 mL of dry DMF was stirred at room temperature for 30 min. Then 5.58 mmol of 2-phenethyl bromide (**2b**) or methyl 4-(bromomethyl)benzoate (**2f**), respectively, were added and the mixture was heated at 80 °C for 4 h. After cooling, the solvent was evaporated and cold H_2_O was added. The suspension was extracted with ethyl acetate (3 × 15 mL) and the collected organic phase was recovered, dried over sodium sulfate and evaporated in vacuum. The crude product was purified by flash column chromatography using cyclohexane/ethyl acetate 1:2 (for 11) or 3:1 (for 12) as eluent.

Ethyl 4-(N,N-diphenethylsulfamoyl)benzoate (**11**). Yield = 12%; mp = 84–87 °C (EtOH). ^1^H-NMR (400 MHz, CDCl_3_) δ 1.41 (t, 3H, COOCH_2_*CH_3_*, *J =* 7.0 Hz), 2.83 (t, 4H, N(CH_2_*CH_2_*Ph)_2_, *J =* 7.8 Hz), 3.40 (t, 4H, N(*CH_2_*CH_2_Ph)_2_, *J =* 7.8 Hz), 4.41 (q, 2H, COO*CH_2_*CH_3_, *J =* 7.0 Hz), 7.13 (d, 4H, Ar, *J =* 7.2 Hz), 7.21–7.30 (m, 6H, Ar), 7.83 (d, 2H, Ar, *J =* 8.0 Hz), 8.13 (d, 2H, Ar, *J =* 8.0 Hz). ESI-MS calcd. for C_25_H_27_NO_4_S, 437.55; found: *m*/*z* 438.17 [M + H]^+^. Anal. C_25_H_27_NO_4_S (C, H, N).

Dimethyl 4,4’-{{{[4-(ethoxycarbonyl)phenyl]sulfonyl}azanediyl}bis(methylene)}dibenzoate (**12**). Yield = 66%; mp = 141–144 °C (EtOH). ^1^H-NMR (400 MHz, DMSO-d_6_) δ 1.45 (t, 3H, COOCH_2_*CH_3_*, *J =* 7.0 Hz), 3.90 (s, 6H, 2 × COO*CH_3_*), 4.38 (s, 4H, N(*CH_2_*PhCOOCH_3_)_2_), 4.45 (q, 2H, COO*CH_2_*CH_3_, *J =* 7.2 Hz), 7.09 (d, 4H, Ar, *J =* 8.4 Hz), 7.88 (d, 4H, Ar, *J =* 8.4 Hz), 7.93 (d, 2H, Ar, *J =* 8.4 Hz), 8.20 (d, 2H, Ar, *J =* 8.4 Hz). ESI-MS calcd. for C_27_H_27_NO_8_S, 525.57; found: *m*/*z* 526.15 [M + H]^+^. Anal. C_27_H_27_NO_8_S (C, H, N).

#### 3.1.9. General Procedure for Synthesis of Compounds **6e** and **13**

A mixture of 0.72 mmol of compounds **11** or **12**, 40% NaOH (19.5 mL) and 0.5 mL of EtOH 96% was stirred at reflux for 2 h. After cooling and maintaining the reaction in ice bath, HCl 6M was added up to acid pH and the precipitate was filtered off to obtain the desired product.

4-(N,N-Diphenethylsulfamoyl)benzoic acid (**6e**). Yield = 42%; mp = 167–170 °C dec. (EtOH). ^1^H-NMR (400 MHz, DMSO-d_6_) δ 2.72 (t, 4H, N(CH_2_*CH_2_*Ph)_2_, *J =* 7.4 Hz), 3.33 (t, 4H, N(*CH_2_*CH_2_Ph)_2_, *J =* 7.4 Hz), 7.25–7.28 (m, 4H, Ar), 7.87 (d, 2H, Ar, *J =* 8.0 Hz), 8.06 (d, 2H, Ar, *J =* 8.0 Hz), 13.35 (exch br s, 1H, COOH). ESI-MS calcd. for C_23_H_23_NO_4_S, 409.50; found: *m*/*z* 410.14 [M + H]^+^. Anal. C_23_H_23_NO_4_S (C, H, N).

4,4′-{{[(4-Carboxyphenyl)sulfonyl]azanediyl}bis(methylene)}dibenzoic acid (**13**). Yield = 99%; mp > 300 °C (EtOH/H_2_O). ^1^H-NMR (400 MHz, DMSO-d_6_) δ 4.41 (s, 4H, N(*CH_2_*PhCOOH)_2_), 7.19 (d, 4H, Ar, *J =* 7.6 Hz), 7.72 (d, 4H, Ar, *J =* 7.6 Hz), 7.99 (d, 2H, Ar, *J =* 8.0 Hz), 8.11 (d, 2H, Ar, *J =* 8.0 Hz). ^13^C-NMR (100 MHz, DMSO-d_6_) δ 52.2 (CH_2_), 127.8 (CH_2_), 128.7 (CH), 124.4 (CH), 129.6 (CH), 130.4 (CH), 130.7 (CH), 135.7 (C), 141.5 (C), 142.8 (C), 166.7 (C), 167.4 (C). ESI-MS calcd. for C_23_H_19_NO_8_S, 469.08; found: *m*/*z* 470.09 [M + H]^+^. Anal. C_23_H_19_NO_8_S (C, H, N).

#### 3.1.10. General Procedure for Synthesis of Compounds **15a**–**c** and **16a**–**c**

A mixture of 0.64 mmol of 5-amino-1-(4-sulfamoylphenyl)-1H-pyrazole-4-carboxylate **14** [33] and 1.28 mmol of anhydrous K_2_CO_3_ in 5 mL of dry DMF was stirred at room temperature for 30 min. Then, 0.96 mmol of appropriate alkyl halide (iodoethane, 1-bromopropane or 1-bromobutane) were added and the mixture was heated at 80 °C for 4 h (for the synthesis of **16a**–**c** an excess (3 equiv.) of alkyl halide was used). After cooling, the solvent was evaporated and the crude product was purified by flash column chromatography using cyclohexane/ethyl acetate 2:1 as eluent.

Ethyl 5-amino-1-(4-(N-ethylsulfamoyl)phenyl)-1H-pyrazole-4-carboxylate (**15a**). Yield = 37%; mp = 166–168 °C (EtOH). ^1^H-NMR (400 MHz, DMSO-d_6_) δ 1.02 (t, 3H, SO_2_NH(CH_2_*CH_3_*), *J =* 7.2 Hz), 1.28 (t, 3H, COOCH_2_*CH_3_*, *J =* 7.0 Hz), 2.83 (quin, 2H, SO_2_NH(*CH_2_*CH_3_), *J =* 7.2 Hz), 4.23 (q, 2H, COO*CH_2_*CH_3_, *J =* 7.0 Hz), 6.56 (exch br s, 2H, NH_2_), 7.68 (t, 1H, SO_2_NH, *J =* 5.8 Hz), 7.77 (s, 1H, pyrazole), 7.80 (d, 2H, Ar, *J =* 8.8 Hz), 7.92 (d, 2H, Ar, *J =* 8.4 Hz). ESI-MS calcd. for C_14_H_18_N_4_O_4_S, 338.10; found: *m*/*z* 339.11 [M + H]^+^. Anal. C_14_H_18_N_4_O_4_S (C, H, N).

Ethyl 5-amino-1-(4-(N-propylsulfamoyl)phenyl)-1H-pyrazole-4-carboxylate (**15b**). Yield = 61%; mp = 123–126 °C (EtOH). ^1^H-NMR (400 MHz, DMSO-d_6_) δ 0.81 (t, 3H, SO_2_NH(CH_2_CH_2_*CH_3_*), *J =* 7.4 Hz), 1.24 (t, 3H, COOCH_2_*CH_3_*, *J =* 7.2 Hz), 1.33–1.42 (m, 2H, SO_2_NH(CH_2_*CH_2_*CH_3_)), 2.71 (q, 2H, SO_2_NH(*CH_2_*CH_2_CH_3_), *J =* 6.8 Hz), 4.19 (q, 2H, COO*CH_2_*CH_3_, *J =* 7.2 Hz), 6.53 (exch br s, 2H, NH_2_), 7.68 (t, 1H, SO_2_*NH*, *J =* 5.6 Hz), 7.76 (d, 3H, 2H Ar + 1H pyrazole, *J =* 8.8 Hz), 7.88 (d, 2H, Ar, *J =* 8.4 Hz). ESI-MS calcd. for C_15_H_20_N_4_O_4_S, 352.12; found: *m*/*z* 353.12 [M + H]^+^. Anal. C_15_H_20_N_4_O_4_S (C, H, N).

Ethyl 5-amino-1-(4-(N-butylsulfamoyl)phenyl)-1H-pyrazole-4-carboxylate (**15c**). Yield = 14%; mp = 130–132 °C (EtOH/H_2_O). ^1^H-NMR (400 MHz, CDCl_3_) δ 0.86 (t, 3H, NHCH_2_CH_2_CH_2_*CH_3_*, *J =* 7.2 Hz), 1.27–1.38 (m, 5H, 3H COOCH_2_*CH_3_* + 2H NHCH_2_CH_2_*CH_2_*CH_3_), 1.45 (quin, 2H, NHCH_2_*CH_2_*CH_2_CH_3_, *J =* 7.2 Hz), 2.97 (t, 2H, NH*CH_2_*CH_2_CH_2_CH_3_, *J =* 7.0 Hz), 4.30 (q, 2H, COO*CH_2_*CH_3_, *J =* 7.0 Hz), 7.74 (d, 2H, Ar, *J =* 8.0 Hz), 7.82 (s, 1H, pyrazole), 7.98 (d, 2H, Ar, *J =* 7.6 Hz). ESI-MS calcd. for C_16_H_22_N_4_O_4_S, 366.14; found: m/z 367.14 [M + H]^+^. Anal. C_16_H_22_N_4_O_4_S (C, H, N).

Ethyl 5-amino-1-(4-(N,N-diethylsulfamoyl)phenyl)-1H-pyrazole-4-carboxylate (**16a**). Yield = 30%; oil. ^1^H-NMR (400 MHz, DMSO-d_6_) δ 1.08 (t, 6H, SO_2_N(CH_2_*CH_3_*)_2_, *J =* 7.2 Hz), 1.28 (t, 3H, COOCH_2_*CH_3_*, *J =* 7.0 Hz), 3.22 (q, 4H, SO_2_N(*CH_2_*CH_3_)_2_, *J =* 7.2 Hz), 4.23 (q, 2H, COO*CH_2_*CH_3_, *J =* 7.0 Hz), 6.58 (exch br s, 2H, NH_2_), 7.80 (d, 3H, 2H Ar + 1H pyrazole, *J =* 6.8 Hz), 7.92 (d, 2H, Ar, *J_1_ =* 6.8 Hz and *J_2_
*= 2.0 Hz). ESI-MS calcd. for C_16_H_22_N_4_O_4_S, 366.14; found: *m*/*z* 367.14 [M + H]^+^. Anal. C_16_H_22_N_4_O_4_S (C, H, N).

Ethyl 5-amino-1-(4-(N,N-dipropylsulfamoyl)phenyl)-1H-pyrazole-4-carboxylate (**16b**). Yield = 14%; oil. ^1^H-NMR (400 MHz, DMSO-d_6_) δ 0.81 (t, 6H, SO_2_N(CH_2_CH_2_*CH_3_*)_2_, *J =* 7.2 Hz), 1.24 (t, 3H, COOCH_2_*CH_3_*, *J =* 7.2 Hz), 1.47 (sex, 4H, SO_2_N(CH_2_*CH_2_*CH_3_)_2_, *J =* 7.6 Hz), 3.03 (t, 4H, SO_2_N(*CH_2_*CH_2_CH_3_)_2_, *J =* 7.6 Hz), 4.19 (q, 2H, COO*CH_2_*CH_3_, *J =* 7.2 Hz), 6.54 (exch br s, 2H, NH_2_), 7.76 (d, 3H, 2H Ar + 1H pyrazole, *J =* 9.2 Hz), 7.89 (d, 2H, Ar, *J =* 8.8 Hz). ESI-MS calcd. for C_18_H_26_N_4_O_4_S, 394.17; found: *m*/*z* 395.17 [M + H]^+^. Anal. C_18_H_26_N_4_O_4_S (C, H, N).

Ethyl 1-(4-(N,N-dipropylsulfamoyl)phenyl)-5-(propylamino)-1H-pyrazole-4-carboxylate (**16c**). Yield = 33%; oil. ^1^H-NMR (400 MHz, DMSO-d_6_) δ 0.66 (t, 3H, NHCH_2_CH_2_*CH_3_*, *J =* 7.4 Hz), 0.77 (t, 6H, SO_2_N(CH_2_CH_2_*CH_3_*)_2_, *J =* 7.4 Hz), 1.24–1.32 (m, 5H, 3H COOCH_2_*CH_3_* + 2H NHCH_2_*CH_2_*CH_3_), 1.43 (quin, 4H, SO_2_N(CH_2_*CH_2_*CH_3_)_2_, *J =* 7.2 Hz), 2.74 (q, 2H, NH*CH_2_*CH_2_CH_3_, *J =* 6.8 Hz), 3.05 (t, 4H, SO_2_N(*CH_2_*CH_2_CH_3_)_2_, *J =* 7.4 Hz), 4.21 (q, 2H, COO*CH_2_*CH_3_, *J =* 7.2 Hz), 6.25 (t, 1H, NH, *J =* 6.2 Hz), 7.74 (d, 2H, Ar, *J =* 8.4 Hz), 7.81 (s, 1H, pyrazole), 7.92 (d, 2H, Ar, *J =* 8.4 Hz). ESI-MS calcd. for C_21_H_32_N_4_O_4_S, 436.21; found: *m*/*z* 437.22 [M + H]^+^. Anal. C_21_H_32_N_4_O_4_S (C, H, N).

#### 3.1.11. General Procedure for Synthesis of Acid Derivatives **17**, **18a**–**c** and **19a**–**c**

A mixture of 0.27 mmol of appropriate ethyl ester derivatives **14**, **15a**–**c** or **16a**–**c**, 0.30 mL of NaOH 6N and 1.50 mL of 96% ethanol was stirred at 100 °C for 4 h. After cooling and maintaining the reaction in ice bath, HCl 6M was added up to acid pH and the precipitate was filtered off to obtain the desired acid products which were purified by crystallisation with ethanol.

5-Amino-1-(4-sulfamoylphenyl)-1H-pyrazole-4-carboxylic acid (**17**). Yield = 48%; mp = 210–211 °C (EtOH). ^1^H-NMR (400 MHz, DMSO-d_6_) δ 6.48 (exch br s, 2H, NH_2_), 7.46 (exh br s, 2H, SO_2_*NH_2_*), 7.71 (s, 1H, pyrazole), 7.74 (d, 2H, Ar, *J =* 8.8 Hz), 7.92 (d, 2H, Ar, *J =* 8.4 Hz), 12.16 (exch br s, 1H, COOH). ^13^C-NMR (100 MHz, DMSO-d_6_) δ 96.2 (C), 123.7 (CH), 127.4 (CH), 141.0 (C), 141.7 (CH), 142.8 (C), 150.8 (C), 165.6 (C). ESI-MS calcd. for C_10_H_10_N_4_O_4_S, 282.04; found: *m*/*z* 283.05 [M + H]^+^. Anal. C_10_H_10_N_4_O_4_S (C, H, N).

5-Amino-1-(4-(N-ethylsulfamoyl)phenyl)-1H-pyrazole-4-carboxylic acid (**18a**). Yield = 75%; mp = 208–210 °C dec. (EtOH). ^1^H-NMR (400 MHz, DMSO-d_6_) δ 1.03 (t, 3H, SO_2_NH(CH_2_*CH_3_*), *J =* 7.2 Hz), 2.83 (quin, 2H, SO_2_NH(*CH_2_*CH_3_), *J =* 7.2 Hz), 6.52 (exch br s, 2H, NH_2_), 7.68 (t, 1H, SO_2_*NH*, *J =* 5.8 Hz), 7.75 (s, 1H, pyrazole), 7.80 (d, 2H, Ar, *J =* 8.8 Hz), 7.92 (d, 2H, Ar, *J =* 8.4 Hz). ^13^C-NMR (100 MHz, DMSO-d_6_) δ 14.1 (CH_3_), 37.4 (CH_2_), 109.0 (C), 123.0 (CH), 127.9 (CH), 140.6 (CH), 142.0 (C), 142.9 (C), 150.2 (C), 164.9 (C). ESI-MS calcd. for C_12_H_14_N_4_O_4_S, 310.07; found: *m*/*z* 311.08 [M + H]^+^. Anal. C_12_H_14_N_4_O_4_S (C, H, N).

5-Amino-1-(4-(N-propylsulfamoyl)phenyl)-1H-pyrazole-4-carboxylic acid (**18b**). Yield = 98%; mp = 215–218 °C (EtOH). ^1^H-NMR (400 MHz, DMSO-d_6_) δ 0.79 (t, 3H, NHCH_2_CH_2_*CH_3_*, *J =* 7.6 Hz), 1.38 (sex, 2H, NHCH_2_*CH_2_*CH_3_, *J =* 7.2 Hz), 2.71 (q, 2H, NH*CH_2_*CH_2_CH_3_, *J =* 6.8 Hz), 6.48 (exch br s, 2H, NH_2_), 7.31 (exch br s, 1H, SO_2_*NH*), 7.72 (s, 1H, pyrazole), 7.76 (d, 2H, Ar, *J =* 8.4 Hz), 7.88 (d, 2H, Ar, *J =* 8.4 Hz). ^13^C-NMR (100 MHz, DMSO-d_6_) δ 11.5 (CH_3_), 22.7 (CH_2_), 44.7 (CH_2_), 124.0 (2 × CH), 128.6 (2 × CH), 139.2 (C), 141.1 (CH), 141.9 (CH), 150.5 (C), 165.5 (C). ESI-MS calcd. for C_13_H_16_N_4_O_4_S, 324.09; found: *m*/*z* 325.09 [M + H]^+^. Anal. C_13_H_16_N_4_O_4_S (C, H, N).

5-Amino-1-(4-(N-butylsulfamoyl)phenyl)-1H-pyrazole-4-carboxylic acid (**18c**). Yield = 92%; mp > 300 °C (EtOH). ^1^H-NMR (400 MHz, DMSO-d_6_) δ 0.78 (t, 3H, NHCH_2_CH_2_CH_2_*CH_3_*, *J =* 7.0 Hz), 1.20–1.26 (m, 2H, NHCH_2_CH_2_*CH_2_*CH_3_), 1.33–1.38 (m, 2H, NHCH_2_*CH_2_*CH_2_CH_3_), 2.74 (q, 2H, NH*CH_2_*CH_2_CH_2_CH_3_, *J =* 6.4 Hz), 6.49 (exch br s, 2H, NH_2_), 7.65 (t, 1H, SO_2_*NH*, *J =* 5.6 Hz), 7.72 (s, 1H, pyrazole), 7.76 (d, 2H, Ar, *J =* 8.4 Hz), 7.88 (d, 2H, Ar, *J =* 8.4 Hz), 12.16 (exch br s, 1H, COOH). ^13^C-NMR (100 MHz, DMSO-d_6_) δ 13.8 (CH_3_), 19.6 (CH_2_), 31.4 (CH_2_), 42.6 (CH_2_), 123.9 (2 × CH), 128.8 (2 × CH), 139.2 (C), 141.3 (C), 141.9 (CH), 150.6 (C), 165.5 (C). ESI-MS calcd. for C_14_H_18_N_4_O_4_S, 338.10; found: *m*/*z* 339.11 [M + H]^+^. Anal. C_14_H_18_N_4_O_4_S (C, H, N).

5-Amino-1-(4-(N,N-diethylsulfamoyl)phenyl)-1H-pyrazole-4-carboxylic acid (**19a**). Yield = 47%; mp = 240 °C dec. (EtOH). ^1^H-NMR (400 MHz, DMSO-d_6_) δ 1.08 (t, 6H, SO_2_N(CH_2_*CH_3_*)_2_, *J =* 7.2 Hz), 3.20 (q, 4H, SO_2_N(*CH_2_*CH_3_)_2_, *J =* 7.2 Hz), 6.52 (exch br s, 2H, NH_2_), 7.70 (s, 1H, pyrazole), 7.82 (d, 2H, Ar, *J =* 8.8 Hz), 7.91 (d, 2H, Ar, *J =* 8.8 Hz). ^13^C-NMR (100 MHz, DMSO-d_6_) δ 10.7 (CH_3_), 38.9 (CH_2_), 109.0 (C), 123.0 (CH), 127.9 (CH), 137.2 (C), 140.6 (CH), 142.9 (C), 150.2 (C), 164.9 (C). ESI-MS calcd. for C_14_H_18_N_4_O_4_S, 338.10; found: *m*/*z* 339.11 [M + H]^+^. Anal. C_14_H_18_N_4_O_4_S (C, H, N).

5-Amino-1-(4-(N,N-dipropylsulfamoyl)phenyl)-1H-pyrazole-4-carboxylic acid (**19b**). Yield = 42%; mp = 167–168 °C (EtOH). ^1^H-NMR (400 MHz, DMSO-d_6_) δ 0.80 (t, 6H, N(CH_2_CH_2_*CH_3_*)_2_, *J =* 7.4 Hz), 1,48 (quin, 4H, N(CH_2_*CH_2_*CH_3_)_2_, *J =* 7.2 Hz), 3.03 (t, 4H, N(*CH_2_*CH_2_CH_3_)_2_, *J =* 7.4 Hz), 6.50 (exch br s, 2H, NH_2_), 7.72 (s, 1H, pyrazole), 7.77 (d, 2H, Ar, *J =* 6.8 Hz), 7.89 (d, 2H, Ar, *J =* 6.8 Hz), 12.17 (exch br s, 1H, COOH). ^13^C-NMR (100 MHz, DMSO-d_6_) δ 11.4 (CH_3_), 22.2 (CH_2_), 50.3 (CH_2_), 123.8 (CH), 128.7 (CH), 138.0 (C), 141.6 (C), 142.0 (CH), 165.6 (C). ESI-MS calcd. for C_16_H_22_N_4_O_4_S, 366.14; found: *m*/*z* 367.14 [M + H]^+^. Anal. C_16_H_22_N_4_O_4_S (C, H, N).

1-(4-(N,N-dipropylsulfamoyl)phenyl)-5-(propylamino)-1H-pyrazole-4-carboxylic acid (**19c**). Yield = 49%; mp = 170–172 °C (EtOH). ^1^H-NMR (400 MHz, DMSO-d_6_) δ 0.66 (t, 3H, NHCH_2_CH_2_*CH_3_*, *J =* 7.4 Hz), 0.77 (t, 6H, SO_2_N(CH_2_CH_2_*CH_3_*)_2_, *J =* 7.2 Hz), 1.28 (quin, 4H, SO_2_N(CH_2_*CH_2_*CH_3_)_2_, *J =* 6.8 Hz), 1.44 (quin, 2H, NHCH_2_*CH_2_*CH_3_, *J =* 7.2 Hz), 2.70–2.75 (m, 2H, NH*CH_2_*CH_2_CH_3_), 3.05 (t, 4H, SO_2_N(*CH_2_*CH_2_CH_3_)_2_, *J =* 7.4 Hz), 6.33 (exch br s, 1H, NH), 7.73–7.78 (m, 3H, 2H Ar + 1H pyrazole), 7.91 (d, 2H, Ar, *J =* 8.4 Hz), 12.23 (exch br s, 1H, COOH). ^13^C-NMR (100 MHz, DMSO-d_6_) δ 11.1 (CH_3_), 11.4 (CH_3_), 21.7 (CH_2_), 23.1 (CH_2_), 47.4 (CH_2_), 49.7 (CH_2_), 125.4 (2 × CH), 128.4 (2 × CH), 139.2 (C), 142.1 (CH), 142.8 (C), 152.7 (C), 167.4 (C). ESI-MS calcd. for C_19_H_28_N_4_O_4_S, 408.18; found: *m*/*z* 409.19 [M + H]^+^. Anal. C_19_H_28_N_4_O_4_S (C, H, N).

#### 3.1.12. General Procedure for Synthesis of Compounds **20**–**22** and **25**–**27**

Compounds **20**–**22** and **25**–**27** were obtained following the same synthetic procedure used for compounds **15a**–**c** and **16a**–**c** but using the appropriate reagent (benzyl bromide for **20**–**22** and **27** and phenethyl bromide for **25** and **26**). The crude compounds were purified by flash chromatography column using cyclohexane/ethyl acetate 2:1 as eluent.

Ethyl 5-amino-1-(4-(N-benzylsulfamoyl)phenyl)-1H-pyrazole-4-carboxylate (**20**). Yield = 23%; oil. ^1^H-NMR (400 MHz, DMSO-d_6_) δ 1.27 (t, 3H, COOCH_2_*CH_3_*, *J =* 7.0 Hz), 4.02 (d, 2H, NH*CH_2_*Ph, *J =* 6.0 Hz), 4.22 (q, 2H, COO*CH_2_*CH_3_, *J =* 7.0 Hz), 6.53 (exch br s, 2H, NH_2_), 7.25–7.30 (m, 5H, Ar), 7.76 (d, 3H, 2H Ar + 1H pyrazole, *J =* 8.0 Hz), 7.93 (d, 2H, Ar, *J =* 8.4 Hz), 8.26 (t, 1H, SO_2_*NH*, *J =* 6.2 Hz). ESI-MS calcd. for C_19_H_20_N_4_O_4_S, 400.12; found: *m*/*z* 401.12 [M + H]^+^. Anal. C_19_H_20_N_4_O_4_S (C, H, N).

Ethyl 5-amino-1-(4-(N,N-dibenzylsulfamoyl)phenyl)-1H-pyrazole-4-carboxylate (**21**). Yield = 42%; oil. ^1^H-NMR (400 MHz, DMSO-d_6_) δ 1.27 (t, 3H, COOCH_2_*CH_3_*, *J =* 6.8 Hz), 4.22 (q, 2H, COO*CH_2_*CH_3_, *J =* 6.8 Hz), 4.33 (s, 4H, N(*CH_2_*Ph)_2_), 6.57 (exch br s, 2H, NH_2_), 7.10–7.36 (m, 10H, Ar), 7.80 (d, 3H, 2H Ar + 1H pyrazole, *J =* 7.2 Hz), 8.00 (d, 2H, Ar, *J =* 8.0 Hz). ESI-MS calcd. for C_26_H_26_N_4_O_4_S, 490.17; found: *m*/*z* 491.17 [M + H]^+^. Anal. C_26_H_26_N_4_O_4_S (C, H, N).

Ethyl 5-(benzylamino)-1-(4-(N,N-dibenzylsulfamoyl)phenyl)-1H-pyrazole-4-carboxylate (**22**). Yield = 10%; oil. ^1^H-NMR (400 MHz, DMSO-d_6_) δ 1.25 (t, 3H, COOCH_2_*CH_3_*, *J =* 7.2 Hz), 3.94 (d, 2H, NH*CH_2_*Ph, *J =* 5.6 Hz), 4.22 (q, 2H, COOCH_2_CH_3_, *J =* 7.2 Hz), 4.35 (s, 4H, SO_2_N(*CH_2_*Ph)_2_), 6.78 (t, 1H, *NH*CH_2_Ph, *J =* 5.6 Hz), 7.10–7.40 (m, 15H, Ar), 7.71 (d, 2H, Ar, *J =* 8.0 Hz), 7.82 (s, 1H, pyrazole), 8.00 (d, 2H, Ar, *J =* 8.0 Hz). ESI-MS calcd. for C_33_H_32_N_4_O_4_S, 580.21; found: *m*/*z* 581.22 [M + H]^+^. Anal. C_33_H_32_N_4_O_4_S (C, H, N).

Ethyl 5-amino-1-(4-(N-phenethylsulfamoyl)phenyl)-1H-pyrazole-4-carboxylate (**25**). Yield = 35%; mp = 108–111 °C (EtOH). ^1^H-NMR (400 MHz, CDCl_3_) δ 1.38 (t, 3H, COOCH_2_*CH_3_*, *J =* 7.0 Hz), 2.81 (t, 2H, NHCH_2_*CH_2_*Ph, *J =* 6.8 Hz), 3.28 (q, 2H, NH*CH_2_*CH_2_Ph, *J =* 6.8 Hz), 4.32 (q, 2H, COO*CH_2_*CH_3_, *J =* 7.0 Hz), 4.48 (t, 1H, SO_2_*NH*, *J =* 6.0 Hz), 5.43 (exch br s, 2H, NH_2_), 7.10 (d, 2H, Ar, *J =* 6.8 Hz), 7.23–7.30 (m, 3H, Ar), 7.50–7.60 (m, 1H, Ar), 7.73 (d, 2H, Ar, *J =* 8.4 Hz), 7.82 (s, 1H, pyrazole), 7.93 (d, 2H, Ar, *J =* 8.4 Hz). ESI-MS calcd. for C_20_H_22_N_4_O_4_S, 414.14; found: *m*/*z* 415.14 [M + H]^+^. Anal. C_20_H_22_N_4_O_4_S (C, H, N).

Ethyl 5-(phenethylamino)-1-(4-(N-phenethylsulfamoyl)phenyl)-1H-pyrazole-4-carboxylate (**26**). Yield = 13%; oil. ^1^H-NMR (400 MHz, CDCl_3_) δ 1.37 (t, 3H, COOCH_2_*CH_3_*, *J =* 7.0 Hz), 2.77 (t, 2H, SO_2_NHCH_2_*CH_2_*Ph, *J =* 7.0 Hz), 2.85–2.90 (m, 2H, NHCH_2_*CH_2_*Ph), 3.41 (t, 2H, NH*CH_2_*CH_2_Ph, *J =* 7.8 Hz), 3.86 (t, 2H, SO_2_NH*CH_2_*CH_2_Ph, *J =* 6.6 Hz), 4.31 (q, 2H, COOCH_2_CH_3_, *J =* 7.0 Hz), 5.47 (exch br s, 1H, NH), 7.15–7.30 (m, 10H, Ar), 7.49 (d, 3H, 2H Ar + 1H pyrazole, *J =* 7.2 Hz), 7.70 (exch br s, 1H, SO_2_*NH*), 7.82 (d, 2H, Ar, *J =* 7.2 Hz). ESI-MS calcd. for C_28_H_30_N_4_O_4_S, 518.20; found: *m*/*z* 519.20 [M + H]^+^. Anal. C_20_H_22_N_4_O_4_S (C, H, N).

Ethyl 5-amino-1-(4-(N-benzyl-N-phenethylsulfamoyl)phenyl)-1H-pyrazole-4-carboxylate (**27**). Yield = 65%; mp = 131–134 °C (EtOH). ^1^H-NMR (400 MHz, DMSO-d_6_) δ 1.26 (t, 3H, COOCH_2_*CH_3_*, *J =* 7.0 Hz), 2.55 (t, 2H, N*CH_2_*CH_2_Ph, *J =* 8.0 Hz), 3.27 (t, 2H, NCH_2_*CH_2_*Ph, *J =* 8.0 Hz), 4.21 (q, 2H, COO*CH_2_*CH_3_, *J =* 7.0 Hz), 4.40 (s, 2H, N*CH_2_*Ph), 6.58 (exch br s, 2H, NH_2_), 6.99 (d, 2H, Ar, *J =* 7.2 Hz), 7.10–7.20 (m, 4H, Ar), 7.30–7.40 (m, 4H, Ar), 7.79 (d, 3H, 2H Ar + 1H pyrazole, *J =* 10.0 Hz), 7.99 (d, 2H, Ar, *J =* 8.0 Hz). ESI-MS calcd. for C_27_H_28_N_4_O_4_S, 504.18; found: *m*/*z* 505.19 [M + H]^+^. Anal. C_27_H_28_N_4_O_4_S (C, H, N).

#### 3.1.13. General Procedure for Synthesis of Acid Derivatives **23**, **24** and **28**–**30**

Compounds **23**, **24** and **28**–**30** were obtained following the same alkaline hydrolysis used for acids **17**, **18a**–**c** and **19a**–**c**. After cooling and maintaining the reaction in ice bath, HCl 6M was added up to acid pH and the precipitate was filtered off to obtain the desired acid products which were purified by crystallisation with ethanol.

5-Amino-1-(4-(N-benzylsulfamoyl)phenyl)-1H-pyrazole-4-carboxylic acid (**23**). Yield = 40%; mp = 207–210 °C (EtOH). ^1^H-NMR (400 MHz, DMSO-d_6_) δ 4.02 (d, 2H, NH*CH_2_*Ph, *J =* 6.0 Hz), 6.48 (exch br s, 2H, NH_2_), 7.25–7.30 (m, 5H, Ar), 7.74 (s, 1H, pyrazole), 7.77 (d, 2H, Ar, *J =* 7.6 Hz), 7.92 (d, 2H, Ar, *J =* 7.6 Hz), 8.25 (t, 1H, *NH*CH_2_Ph, *J =* 6.0 Hz), 12.14 (exch br s, 1H, COOH). ^13^C-NMR (100 MHz, DMSO-d_6_) δ 46.2 (CH_2_), 109.0 (CH), 123.0 (CH), 126.7 (CH), 126.9 (CH), 127.9 (CH), 128.5 (CH), 140.6 (CH), 141.6 (CH), 142.2 (C), 142.9 (C), 150.2 (C), 164.9 (C). ESI-MS calcd. for C_17_H_16_N_4_O_4_S, 372.09; found: *m*/*z* 373.09 [M + H]^+^. Anal. C_17_H_16_N_4_O_4_S (C, H, N).

5-Amino-1-(4-(N,N-dibenzylsulfamoyl)phenyl)-1H-pyrazole-4-carboxylic acid (**24**). Yield = 20%; mp = 168–170 °C (EtOH). ^1^H-NMR (400 MHz, DMSO-d_6_) δ 4.32 (s, 4H, SO_2_N(*CH_2_*Ph)_2_), 6.54 (exch br s, 2H, NH_2_), 7.05–7.15 (m, 4H, Ar), 7.19–7.25 (m, 6H, Ar), 7.72 (s, 1H, pyrazole), 7.83 (d, 2H, Ar, *J =* 8.4 Hz), 7.99 (d, 2H, Ar, *J =* 8.4 Hz). ^13^C-NMR (100 MHz, DMSO-d_6_) δ 51.9 (CH_2_), 109.0 (C), 123.3 (CH), 127.9 (CH), 128.7 (CH), 128.9 (CH), 136.6 (C), 137.4 (C), 140.6 (CH), 142.9 (C), 150.5 (C), 164.9 (C). ESI-MS calcd. for C_24_H_22_N_4_O_4_S, 462.14; found: *m*/*z* 463.14 [M + H]^+^. Anal. C_24_H_22_N_4_O_4_S (C, H, N).

5-Amino-1-(4-(N-phenethylsulfamoyl)phenyl)-1H-pyrazole-4-carboxylic acid (**28**). Yield = 13%; mp = 186–188 °C (EtOH). ^1^H-NMR (400 MHz, DMSO-d_6_) δ 2.71 (t, 2H, NHCH_2_*CH_2_*Ph, *J =* 7.4 Hz), 3.00 (q, 2H, NH*CH_2_*CH_2_Ph, *J =* 7.0 Hz), 6.49 (exch br s, 2H, NH_2_), 7.15–7.20 (m, 3H, Ar), 7.26 (t, 2H, Ar, *J =* 7.4 Hz), 7.73 (s, 1H, pyrazole), 7.77 (d, 2H, Ar, *J =* 8.8 Hz), 7.82 (t, 1H, SO_2_*NH*, *J =* 5.8 Hz), 7.89 (d, 2H, Ar, *J =* 8.8 Hz). ^13^C-NMR (100 MHz, DMSO-d_6_) δ 35.7 (CH_2_), 42.4 (CH_2_), 109.0 (C), 123.8 (CH), 126.7 (CH), 128.4 (CH), 128.8 (CH), 129.1 (CH), 139.1 (C), 141.4 (C), 141.9 (CH), 150.2 (C), 164.9 (C). ESI-MS calcd. for C_18_H_18_N_4_O_4_S, 386.10; found: *m*/*z* 387.11 [M + H]^+^. Anal. C_18_H_18_N_4_O_4_S (C, H, N).

5-(Phenethylamino)-1-(4-(N-phenethylsulfamoyl)phenyl)-1H-pyrazole-4-carboxylic acid (**29**). Yield = 18%; mp = 105–107 °C (EtOH). ^1^H-NMR (400 MHz, DMSO-d_6_) δ 2.65 (t, 2H, NHCH_2_*CH_2_*Ph, *J =* 7.2 Hz), 2.77 (t, 2H, NH*CH_2_*CH_2_Ph, *J =* 7.2 Hz), 2.94 (t, 2H, SO_2_NHCH_2_*CH_2_*Ph, *J =* 6.4 Hz), 3.35 (t, 2H, SO_2_NH*CH_2_*CH_2_Ph, *J =* 7.6 Hz), 6.50 (exch br s, 1H, *NH*CH_2_CH_2_Ph), 7.10–7.30 (m, 10H, Ar), 7.55–7.62 (m, 3H, 2H Ar + 1H pyrazole), 7.70 (exch br s, 1H, SO_2_*NH*), 7.81 (d, 2H, Ar, *J =* 8.4 Hz). ^13^C-NMR (100 MHz, DMSO-d_6_) δ 35.3 (CH_2_), 35.6 (CH_2_), 44.4 (CH_2_), 50.9 (CH_2_), 109.0 (C), 123.1 (CH), 126.7 (CH), 126.9 (CH), 127.9 (C), 128.8 (CH), 128.9 (CH), 129.1 (CH), 129.2 (CH), 129.7 (CH), 132.9 (CH), 139.4 (C), 142.0 (C), 142.9 (C), 151.2 (C), 164.9 (C). ESI-MS calcd. for C_26_H_26_N_4_O_4_S, 490.17; found: *m*/*z* 491.17 [M + H]^+^. Anal. C_26_H_26_N_4_O_4_S, (C, H, N).

5-Amino-1-(4-(N-benzyl-N-phenethylsulfamoyl)phenyl)-1H-pyrazole-4-carboxylic acid (**30**). Yield = 59%; mp = 263–266 °C dec. (EtOH). ^1^H-NMR (400 MHz, DMSO-d_6_) δ 2.55 (t, 2H,N*CH_2_*CH_2_Ph, *J =* 7.4 Hz), 3.26 (t, 2H, NHCH_2_*CH_2_*Ph, *J =* 7.6 Hz), 4.39 (s, 2H,N*CH_2_*Ph), 6.53 (exch br s, 2H, NH_2_), 6.99 (d, 2H, Ar, *J =* 7.2 Hz), 7.15–7.20 (m, 3H, Ar), 7.31–7.36 (m, 5H, Ar), 7.61 (s, 1H, pyrazole), 7.86 (d, 2H, Ar, *J =* 8.0 Hz), 7.95 (d, 2H, Ar, *J =* 8.4 Hz). ^13^C-NMR (100 MHz, DMSO-d_6_) δ 34.9 (CH_2_), 49.9 (CH_2_), 52.0 (CH_2_), 103.4 (C), 122.6 (CH), 126.8 (CH), 128.2 (CH), 128.7 (C), 128.9 (CH), 128.9 (CH), 129.0 (CH), 136.6 (C), 137.2 (C), 138.6 (C), 142.8 (CH), 142.9 (C), 149.7 (C), 168.4 (C). ESI-MS calcd. for C_25_H_24_N_4_O_4_S, 476.15; found: *m*/*z* 477.16 [M + H]^+^. Anal. C_25_H_24_N_4_O_4_S (C, H, N).

### 3.2. Pharmacological Assays

#### 3.2.1. Preparation of Oocytes

Oocytes of *Xenopus laevis* are prepared as previously described in [39] where the oocytes are isolated from surgically removed ovary segments incubated in an Oocyte Ringe solution Ca_2_-free (OR2: in mM: 82.5 NaCl, 2.5 KCl, 1.0 MgCl_2_, 1.0 CaCl_2_, 1.0 Na_2_HPO_4_ e 5.0 HEPES, pH 7.5) with 2 mg/mL of collagenase and antibiotics (10,000 U/mL penicillin and 10 mg/mL of streptomycin) stirred at room temperature for 3 h. After being washed with OR2, oocytes free of follicular cells and that have a uniform pigmentation are chosen and stored in OR2 for 18 °C.

#### 3.2.2. Synthesis of mRNA

Mouse pannexin1, in pCS2, was linearised with NotI. In vitro transcription was performed with the polymerase SP6, using the Message Machine kit (Ambion). mRNA was quantified by absorbance (260 nm), and the proportion of full-length transcripts was checked by agarose gel electrophoresis. In vitro-transcribed mRNA (~20 nL) was injected into *Xenopus laevis* oocytes.

#### 3.2.3. Electrophysiology

Whole-cell membrane currents of oocytes were measured using a two-electrode voltage clamp (Gene Clamp 500B, Axon Instruments/Molecular Devices Sunnyvale, Union City, CA, USA). Glass pipettes were pulled using a P-97 Flaming/Brown type puller (Sutter, Novato, CA, USA). The recording chamber was perfused continuously with frog Ringer (OR) solution (in mM: 82.5 NaCl, 2.5 KCl, 1 CaCl2, 1 MgCl2, 1 Na2HPO4, and 5 HEPES, pH 7.5).

Membrane conductance was determined using voltage pulses, and the pulse-induced current amplitudes were divided by the amplitudes of the voltage steps. For calculation of % inhibition, leak currents were subtracted. Typically, leak currents after the interventions were smaller than at the beginning of the experiment and this smaller value was used for the subtraction to avoid overestimation of the inhibitory effect. For an independent measure of the leak current, oocytes at the end of the experiment were exposed to 100 µM carbenoxolone, which is known to close Panx1 channels 100%. Oocytes expressing Panx1 were held at −60 mV, and pulses to +60 mV were applied to transiently open the channels by means of the voltage gate. Pulses 5 s in duration were applied at 0.1 Hz for current and conductance measurements.

### 3.3. In Vivo Tests

#### 3.3.1. Study Approval

All animal manipulations were carried out according to the Directive 2010/63/EU of the European parliament and of the European Union council (22 September 2010) on the protection of animals used for scientific purposes and with IASP. The ethical policy of the University of Florence complies with the Guide for the Care and Use of Laboratory Animals of the US National Institutes of Health (NIH Publication No. 85-23, revised 1996; University of Florence assurance number: A5278-01). Formal approval to conduct the experiments described was obtained from the Italian Ministry of Health (No. 171/2018-PR) and from the Animal Subjects Review Board of the University of Florence and from the Animal Ethics Committee of University of Campania of Naples. Experiments involving animals have been reported according to ARRIVE guidelines [40] All efforts were made to minimise animal suffering and to reduce the number of animals used.

#### 3.3.2. Animals

Eight-week-old male CD-1 mice (Envigo, Varese, Italy) weighing approximately 20–25 g at the beginning of the experimental procedure were used. Animals were housed in the Centro Stabulazione Animali da Laboratorio (CeSAL; University of Florence, Italy) and used at least 1 week after their arrival; 10 mice were housed per cage (size 26 cm × 41 cm). Animals were fed a standard laboratory diet and tap water ad libitum and kept at 23 ± 1 °C with a 12 h light/dark cycle (light at 7 a.m.).

#### 3.3.3. Oxaliplatin-Induced Neuropathic Pain Model

Mice treated with oxaliplatin (Carbosynth, Pangbourne, UK; 2.4 mg kg^−1^) were administered intraperitoneally (i.p.) for 2 weeks [16,41] Oxaliplatin was dissolved in a 5% glucose solution. Control animals received an equivalent volume of vehicle. Behavioural tests were performed starting from day 14, when neuropathy was well established.

#### 3.3.4. Compounds Administration

Compounds **4** and **13** were intrathecally (i.t.) injected in conscious mice as previously described [42]. Compound **4** was dissolved in sterile saline solution while compound **13** was dissolved in a mixture consisting of PEG400, TWEEN 20 and sterile saline solution (1:1:3, respectively). Briefly, a 25 µL Hamilton syringe connected to a 30-gauge needle was intervertebrally inserted between the L4 and L5 region, and advanced 6 mm into the lumbar enlargement of the spinal cord. Behavioural measurements were performed before and 15 min, 30 min, 45 min, and 60 min after the administration of compounds.

#### 3.3.5. Assessment of Thermal Allodynia (Cold Plate Test)

Thermal allodynia was assessed using the Cold-plate test. With minimal animal-handler interaction, mice were taken from home-cages, and placed onto the surface of the cold-plate (Ugo Basile, Varese, Italy) maintained at a constant temperature of 4 ± 1 °C. Ambulation was restricted by a cylindrical plexiglas chamber (diameter: 10 cm, height: 15 cm), with open top. A timer controlled by foot peddle began timing response latency from the moment the mouse was placed onto the cold-plate. Pain-related behaviour (licking of the hind paw) was observed, and the time (seconds) of the first sign was recorded. The cut-off time of the latency of paw lifting or licking was set at 30 s [43].

#### 3.3.6. Statistics

Results were expressed as means ± SEM and the analysis of variance was performed by ANOVA test. A Bonferroni’s significant difference procedure was used as post hoc comparison. P values less than 0.05 were considered significant. Data were analysed using “Origin^®^ 10” software.

### 3.4. Molecular Dynamics and Statistical Analysis

The structure of the channel was obtained from the hPANX-1 (PDB ID 7DWB), considering all the amino acids within a distance of about 25 Å from the centre of the pore, identified as ‘CBX binding site’. The ligands were placed at the binding site through AUTODOCK 4.2 [36]. The main autodock parameters used in this work are in Table 7 and Table 8:

The molecular dynamics simulations of ligand binding-site complexes were performed on a minimum number of conformations (maximum 2) such as to cover at least 70% of the poses found by AUTODOCK.

A 10 ns MD simulation was performed for all complexes using GROMACS v5.1 program, and it was conducted in vacuum [37]. The DS ViewerPro 6.0 program [44] was used to build the initial conformations of ligands. The partial atomic charge of the ligand structures was calculated with CHIMERA [45] using AM1-BCC method, and the topology was created with ACPYPE [46] based on the routine Antechamber [47].

The OPLS-AA/L all-atom force field [48] parameters were applied to all the structures. To remove bad contacts, the energy minimisation was performed using the steepest descent algorithm until convergence is achieved or for 50,000 maximum steps. The next equilibration of the system was conducted in two phases:(1)canonical NVT ensemble, a 100 ps position-restrained of molecules at 300 K was carried out using a temperature-coupling thermostat (velocity rescaling with a stochastic term) to ensure the proper stabilisation of the temperature [49];(2)isothermal isobaric NPT ensemble, a 100 ps position-restrained of molecules at 300 K and 1 bar was carried out without using barostat pressure coupling to stabilise the system. These were then followed by a 10 ns MD run at 300 K with position restraints for all protein atoms. The Lincs algorithm [50] was used for bond constraints to maintain rigid bond lengths.

The initial velocity was randomly assigned taken from Maxwell–Boltzman distribution at 300 K and computed with a time step of 2 fs, and the coordinates were recorded every 0.1 ns. The conformations collected during the simulated trajectory were 100.

The Proximity Frequencies (PFs), with which the 100 conformations of each binding-site ligand complex intercepts two or more amino acid during the dynamic simulation, were calculated. The statistical analysis was carried out with OriginPro2018 program [51].

## 4. Conclusions

In a previous work we reported the first series of de-novo Panx-1 channel blockers, obtained following a rational approach [23]; in the present paper we describe the further development of this research, which was based on the information gained from the results of the first series, thus taking into account the structural requirements for the activity. Therefore, we went to include on the naphthalene and pyrazole nucleus those functions and groups that in the first series were favourable for activity, in addition to other groups reported in the literature. The active compounds are characterised by a lipophilic scaffold with a certain steric hindrance into which one or more acid groups are inserted, as well as by the presence of a sulphonamide function, appropriately substituted. These data are in agreement with our previous results. Compounds **4** and **13**, which are an analogue of the drug Probenecid, are the most potent Panx-1 inhibitors obtained, able to almost completely block the channel opening at 50 µM (**4**, I = 97% and **13**, I = 93.7%). Both compounds were tested intrathecally in a mouse model of oxaliplatin-induced neuropathic pain at the dose of 1 nmol and 3 nmol; they had a similar profile showing the maximum anti-allodynic effect on 45 min with the dose of 3 nmol. These results suggested a relationship between this effect and the channel blocking ability of compounds 4 and 13 and indicated the class of Panx-1 inhibitors as an interesting new approach for chemotherapy-induced neuropathic pain.

Finally, from the MD study and the PCA analysis, it has been possible to individuate a discriminating factor separating active compounds from inactive ones. Considering that seven identical chains form the channel, ligands could occupy the same position in each protomer making it difficult the recognise a binding site for Panx-1 inhibitory ligands. Thus, evaluating the number of PFs represents the only tool we could use as the starting point in the study of Panx-1 inhibitors. The PFs collected for the inactive compounds are lower than the active compounds. The different distribution in the PCA graph of compounds (the inactive compounds are grouped in a ‘statistically significant manner’, i.e., nearly zero interactions) indicate that our approach could be helpful in the study of Panx-1 channel inhibitors.

## Data Availability

The data that support the findings of this study are available from the corresponding author upon reasonable request.

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
