# Peer review of "New Panx-1 Blockers: Synthesis, Biological Evaluation and Molecular Dynamic Studies"

_ijms, 2022, doi:10.3390/ijms23094827_

Round 1

Reviewer 1 Report

In general, this work is really interesting and well written. My comments aim to increase the scientific soundness and clarity of it.

Line 187 – carbenoxolone is abbreviated to CBX. However, the first appearance of CBX is in line 67. Moreover, in line 84 there is also not abbreviated carbenoxolone. Please correct this.

Figures – why references to figures are written bold? Does it have any special meaning?

Figure 1 caption could be corrected. Sounds awkward. Maybe “Examples of Panx-1 blockers shown and described in the scientific literature”.

Figure 2 – please correct the figure 2 caption. It is not informative enough what A, B and C stands for. Please also use (Figure 2A) instead of (A, figure 2).

Line 121 – please explain what DMF stands for.

Lines 90, 258 – please expand “NPPB”

Line 258 - “BB FCF” should be changed to Brilliant Blue FCF (consequently to Figure 1)

Lines 184, 222, 227, 236 – “Xenopus Laevis” should be written in italics

Line 863 – It is not clear how many were mice treated with oxaliplatin (n=?) and how many were control animals (n=?).

Line 865, 869 – why “i.p.” is not explained whereas “i.t.” is?

Author Response

In general, this work is really interesting and well written. My comments aim to increase the scientific soundness and clarity of it.

We thank the Reviewer for the comments and we have done all the suggested modifications. In particular:

 Line 187 – carbenoxolone is abbreviated to CBX. However, the first appearance of CBX is in line 67. Moreover, in line 84 there is also not abbreviated carbenoxolone. Please correct this.

Done. It has also been specified in Figure 1.

Figures – why references to figures are written bold? Does it have any special meaning?

The authors have bolded the word 'Table' in the text to make the reference immediately usable by the reader. In any case, if this is not correct, we can remove the bold.

Figure 1 caption could be corrected. Sounds awkward. Maybe “Examples of Panx-1 blockers shown and described in the scientific literature”.

Done

Figure 2 – please correct the figure 2 caption. It is not informative enough what A, B and C stands for. Please also use (Figure 2A) instead of (A, figure 2).

Done

Line 121 – please explain what DMF stands for.

Done

Lines 90, 258 – please expand “NPPB”

Done. It has also been specified in Figure 1.

Line 258 - “BB FCF” should be changed to Brilliant Blue FCF (consequently to Figure 1)

Done (see Table 4)

Lines 184, 222, 227, 236 – “Xenopus Laevis” should be written in italics

Done

Line 863 – It is not clear how many were mice treated with oxaliplatin (n=?) and how many were control animals (n=?).

As stated in the Figure 5 legend "Each value represents the mean ± S.E.M. of 10 mice performed in two different experimental sets", therefore all groups of treatment are populated by 10 mice (10 treated with oxaliplatin and 10 control animals), and experiments were performed in two sessions. This last phrase has been inserted in the legend.

Line 865, 869 – why “i.p.” is not explained whereas “i.t.” is?

We have specified “i.p.” (intraperitoneal) in Figure 5 and in the experimental section.

Reviewer 2 Report

In this report Dahl et al. reported design and synthesis of a series of PanX-1 blocker. A series of molecules were synthesized and studied. Although the paper can not be accepted in its current form. The paper can be reconsidered after the following changes.

  1. The molecular characterization need to be improved before next submission. In many 13C spectrum, the peaks are inverted. What is the reason for that?
  2. For any medicinal studies molecular purity is an important requirement. In this regard for each studied molecule LCMS or HPLC data need to be included.
  3. The IC50 values are in the high range for these molecules. The authors need to comment on this.

Author Response

Reviewer 2

In this report Dahl et al. reported design and synthesis of a series of PanX-1 blocker. A series of molecules were synthesized and studied. Although the paper can not be accepted in its current form. The paper can be reconsidered after the following changes.

We thank the Reviewer for the comments and below are the answers to the remarks. In particular:

  1. The molecular characterization need to be improved before next submission. In many 13C spectrum, the peaks are inverted. What is the reason for that?

We are sorry for this problem and the reason is that the phase has been 'mistakenly' reversed by the operator, but this inversion does not affect the interpretation of the spectrum. In any case, we re-registered the spectra in the right way and replaced the inverted ones with the new corrected ones in the Supporting Information.

  1. For any medicinal studies molecular purity is an important requirement. In this regard for each studied molecule LCMS or HPLC data need to be included.

To assess the purity of compounds, the elemental analyses for carbon, hydrogen and nitrogen is traditionally deemed acceptable if the accuracy of the results is within ± 0.4% of the theoretical values for the proposed formula. Remaining the values of the elemental analysis of all our compounds in the range ± 0.4, we can state with reasonable confidence that our products meet the purity requirements. Moreover, we included ESI-MS results as additional data for assessing purity of molecules. LCMS and HPLC data would certainly be the most suitable for defining the purity of the final compounds, but unfortunately at this time we do not have the possibility to obtain them. We hope that the Reviewer will agree that elemental analysis and ESI-MS are sufficient to define the purity of the final products.

  1. The IC50 values are in the high range for these molecules. The authors need to comment on this.

The IC50 values reported here range from 1.2 to 40.4 µM and thus are in the same range as (and one even lower than) the reference compounds CBX, NPPB and probenecid. In the literature, Probenecid, with an IC50 of 150 µM, is one of the most commonly used drugs to test involvement of Panx1 in biological functions. CBX has an IC50 of 10 µM on Panx1. The most efficacious inhibitor of Panx1 presently known is the food dye Brilliant Blue For Coloring Food (BB FCF). However, like Brilliant Blue G, BB FCF is a strong dye and a clinical use would involve tinting skin and sclera blue. This certainly would limit acceptance of BB FCF as a therapeutic by the general public.

In this respect, we have included an explanatory sentence at the end of the paragraph "Biological evaluation”.
